# Mutual Reasoning Makes Smaller LLMs Stronger Problem-Solvers

**Zhenting Qi**[*‡†] **Mingyuan Ma**[*‡†] **Jiahang Xu**[*‡] **Li Lyna Zhang**[‡◇] **Fan Yang**[‡] **Mao Yang**[‡]

[‡]Microsoft Research Asia [†]Harvard University

## Abstract

This paper introduces rStar, a self-play mutual reasoning approach that significantly improves reasoning capabilities of small language models (SLMs) without fine-tuning or superior models. rStar decouples reasoning into a self-play mutual generation-discrimination process. First, a target SLM augments the Monte Carlo Tree Search (MCTS) with *a rich set of human-like reasoning actions* to construct higher quality reasoning trajectories. Next, another SLM, with capabilities similar to the target SLM, acts as a discriminator to verify each trajectory generated by the target SLM. The mutually agreed reasoning trajectories are considered *mutual consistent*, thus are more likely to be correct. Extensive experiments across five SLMs demonstrate rStar can effectively solve diverse reasoning problems, including GSM8K, GSM-Hard, MATH, SVAMP, and StrategyQA. Remarkably, rStar boosts GSM8K accuracy from 12.51% to 63.91% for LLaMA2-7B, from 36.46% to 81.88% for Mistral, from 74.53% to 91.13% for LLaMA3-8B-Instruct. Code is available at https://github.com/zhentingqi/rStar.

## 1 Introduction

Despite their success, large language models (LLMs) face significant challenges in complex reasoning (Valmeekam et al., 2022; Weng et al., 2023). For example, state of the art models like Mistral-7B (Jiang et al., 2023) can only achieve 36.5% accuracy on the GSM8K dataset, even with techniques like Chain-of-Throught (CoT) (Wei et al., 2022). Although fine-tuning is shown to be an effective way to improve reasoning capability, most LLMs rely on fine-tuning data distilled or synthesized by *superior* models like GPT-4 (Wang et al., 2024a; Gou et al., 2023). Meanwhile, the community has been actively working on a complimentary and yet more challenging approach: Reasoning improvements *without* a superior teacher LLM.

A promising paradigm to improve reasoning without superior models is to leverage the knowledge within LLMs themselves (Wang et al., 2023; Hao et al., 2023; Madaan et al., 2024). For example, RAP (Hao et al., 2023) adopts a self-exploration solution to iteratively improve LLM's reasoning performance through self-rewarded feedback. Unfortunately, study suggests that this paradigm often suffers from two fundamental issues.

First, LLMs often struggle to effectively explore the solution space during reasoning. The self-exploration often traps in a solution space with low-quality reasoning steps even after many attempts. For example, our experiments reveal that after 32 rounds of self-exploration with RAP (Hao et al., 2023), only 24% of the trajectories generated by LLaMA2-7B on GSM8K are correct. Second, even the self-exploration can find high quality reasoning steps, it is difficult for

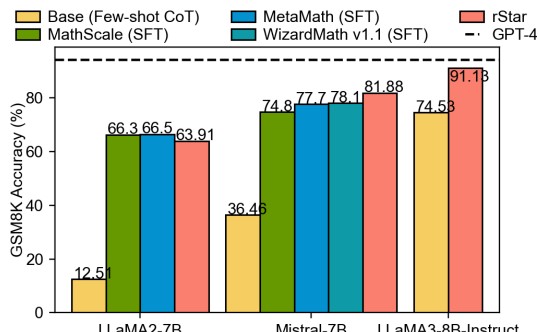

Figure 1: With 32 rounds of inference, rStar makes SLMs highly capable problem-solvers, matching or even surpassing the reasoning performance achieved after domain-specialized SFT.

---

[*]Equal contribution. Zhenting Qi and Mingyuan Ma did the work during an internship at MSRA
[◇]Correspondence to: lzhani@microsoft.com

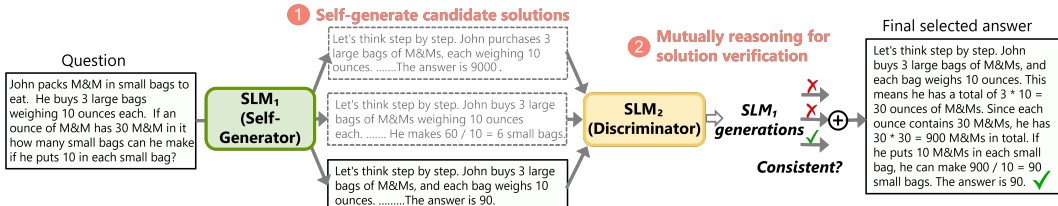

Figure 2: Our self-play mutual reasoning is a generation-discrimination process: (1) a self-generator augments the target SLM to generate candidate reasoning trajectories using MCTS; (2) the discriminator uses another SLM to provide unsupervised feedback on each trajectory based on partial hints; (3) based on this feedback, the target SLM decides a final reasoning trajectory as the solution.

LLMs to tell which reasoning steps are of higher quality or determine which final answers are correct, thus it is hard to effectively guide the self-exploration. Our study shows that a naïve reward-based self-exploration guidance can lead to results no better than random guesses (see Appendix A.1).

A more troublesome fact is that the above two issues are more pronounced in the smaller version of LLMs, i.e., **SLM**s, due to their weaker capabilities. For instance, while GPT-4 can improve by self-refining its output (Madaan et al., 2024; Wu et al., 2024; Zhou et al., 2024), the approaches are less effective in SLMs and may even lead to worse performance (Forsman, 2024). This significantly hinders the adoption of neural language models.

This paper introduces *Self-play muTuAl Reasoning* (rStar), a novel approach that boosts SLMs' reasoning capability during inference without fine-tuning or superior models. To address the aforementioned challenges, rStar decouples reasoning into a self-play mutual generation-discrimination process as illustrated in Fig. 2. Specifically, rStar is unique in the following approaches. First, although relying on a conventional Monte Carlo Tree Search (MCTS) for SLMs to self-generate reasoning steps, rStar advocates *a richer set of reasoning actions* in the self-exploration. The new proposed actions simulate human reasoning behaviors given the current reasoning state, such as decomposing and searching for a specific reasoning step, proposing a new sub-question, or rephrasing the given question. This enables SLMs to generate high-quality candidate reasoning trajectories during self-exploration. Second, to effectively guide the exploration among the generated reasoning trajectories, rStar augments the MCTS process with a new discrimination process called *mutual consistency*. In particular, rStar employs a second SLM with the similar capability, acting as a discriminator to provide unsupervised feedback on each candidate reasoning trajectory generated by MCTS. To improve the accuracy of the feedback, rStar hints the second SLM with sampled partial reasoning trajectories, asking it to complete the remaining reasoning steps. And rStar deems the mutually agreed reasoning trajectories of higher quality. Mutual consistency mirrors the common human practice in the absence of supervision, where agreement among peers (i.e., two SLMs) on derived answers suggests a higher likelihood of correctness. As a result, mutual consistency offers more effective reasoning across diverse tasks than other approaches like self-consistency (Wang et al., 2023) and avoids the risk of overfitting when training a reward model (Chen et al., 2024a; Wang et al., 2024b).

Extensive experiments across five SLMs and five diverse reasoning tasks demonstrate the effectiveness of rStar. With just 32 rounds of MCTS inference, rStar significantly enhances SLMs' reasoning capabilities, matching or even surpassing the accuracy achieved after fine-tuning. For example, rStar boosts GSM8K accuracy from 12.51% to 63.91% for LLaMA2-7B, from 36.46% to 81.88% for Mistral, and from 47.23% to 85.52% for LLaMA3-8B. Furthermore, we conduct comprehensive experiments to verify rStar's superiority over state-of-the-art baselines, including single-round inference techniques like few-shot CoT, multi-round prompting approaches such as self-consistency, and self-improvement techniques such as RAP, ToT, self-evaluation and self-verification.

## 2  RELATED WORK

**Prompting Language Models to Reason**. Prompting-based methods, such as Chain-of-Thought (Wei et al., 2022), focus on designing instructions and pipelines to enhance LLMs' reasoning performance during inference. Recent advances include planning (Hao et al., 2023; Ding et al., 2023), problem decomposition (Zhou et al., 2022; Khot et al., 2022; Hao et al., 2023), abstraction (Zheng et al., 2023),

programming (Chen et al., 2022; Zhou et al., 2023). These methods aim to improve single-round inference performance and are orthogonal to ours.

**LLM Self-improvement**. Recently, research on the self-improvement of LLMs has rapidly increased. Fine-tuning based methods (Chen et al., 2024b;a) leverage the capabilities of a well-pretrained LLM to synthesize data and progressively enhance its performance. Advanced prompting techniques, such as self-verification (Gero et al., 2023; Zhou et al., 2023), and RAP (Hao et al., 2023), improve performance through iterative self-exploring based on self-diagnosed feedback at inference time. However, as illustrated in previous section, the achieved performance often depend on the LLM's inherent capabilities, and for SLMs, their weaker instruction-following ability and unreliable self-rewarding can mislead self-improvement.

**Sampling Reasoning Paths**. Recent works (Brown et al., 2024; Li et al., 2024; Snell et al., 2024) on mathematical reasoning have shown that sampling diverse reasoning paths can significantly enhance performance compared to greedy one-time decoding. Self-Consistency (Wang et al., 2023) sample a complete CoT path each time. Tree-search approaches (Yao et al., 2024; Hao et al., 2023; Zhang et al., 2024), like MCTS, further improve the performance by breaking down tasks and sampling simpler, individual intermediate reasoning steps. However, most approaches have limited action spaces. For example, RAP (Hao et al., 2023) decomposes only subproblems, while AlphaMath (Chen et al., 2024a) searches only for one CoT step, limiting effectiveness in generating better trajectories.

**Answer Verification**. To select correct reasoning trajectories, majority voting (Wang et al., 2023) is a widely-used approach. To improve accuracy, some works train value or rewards model for verification (Wang et al., 2024b; Chen et al., 2024a), but these require additional annotations and have risks in overfitting to specific tasks. Self-verification (Weng et al., 2023) leverages LLM capabilities for backward self-verification. Nevertheless, its effectiveness hinges on its inherent ability to reason effectively. Recent studies have shown that LLM struggles to evaluate itself and rectify its initial responses without any external feedbacks (Huang et al., 2023; Feng et al., 2023).

# 3 METHODOLOGY

## 3.1 OVERVIEW

**Problem Formulation**. To solve a reasoning problem by SLMs, we formulate it as a multi-step reasoning generation task, which breaks the problem into simpler sub-tasks. This is more effective than traditional CoT-based reasoning (Wei et al., 2022; Wang et al., 2023), as it is much easier for SLMs to correctly generate one step than complete reasoning steps in a single inference. We leverage the Monte-Carlo Tree Search (MCTS) algorithm (Kocsis & Szepesvári, 2006) to augment the target SLM for self-generating multi-step reasoning solutions.

Formally, for a given problem $x$ and a target SLM $M$, the MCTS augments $M$ to incrementally build a search tree $\mathcal{T}$. As illustrated in Fig. 3, the root node represents the question $x$, an edge represents an action $a$, each child node is an intermediate step $s$ generated by $M$ under the corresponding action. A path from the root node to a leaf node (denoted as $s_d$, also called a terminal node) constitutes a candidate solution trajectory $\mathbf{t} = x \oplus s_1 \oplus s_2 \oplus ... \oplus s_d$. From the search tree $\mathcal{T}$, we can extract a set of solution trajectories $\mathbb{T} = \{\mathbf{t}^1, \mathbf{t}^2, ..., \mathbf{t}^n\}(n \geq 1)$. Our goal is to find the trajectories that can achieve the correct answer for the given question.

**Challenges in SLM Self-Improvement**. MCTS allows an SLM to explore and evaluate multiple potential solutions. Ideally, by balancing exploration of new possibilities with the exploitation of high-reward actions, the SLM can gradually refine its reasoning steps to generate a final correct reasoning trajectory. However, due to the limited capabilities in SLMs, traditional MCTS yields minimal improvement. First, the vast solution space makes it difficult for SLMs to generate effective solutions. Existing MCTS-based methods (Hao et al., 2023; Kang et al., 2024) relying on single actions limit diversity and struggle to generalize across tasks. While self-consistency (Wang et al., 2023) can ensure diversity through random sampling, SLMs often generate poor-quality solutions, requiring many attempts to find a correct solution and increasing inference costs.

Second, it's challenging to accurately reward each action. Without ground truth labels, it's difficult to verify the correctness for each intermediate step $s_i$ and the final answer in $s_d$. Majority voting in self-consistency requires most traces to be correct, which is often not the case for SLMs. Methods like RAP (Hao et al., 2023) use self-rewarding, but our study shows SLMs perform near-random

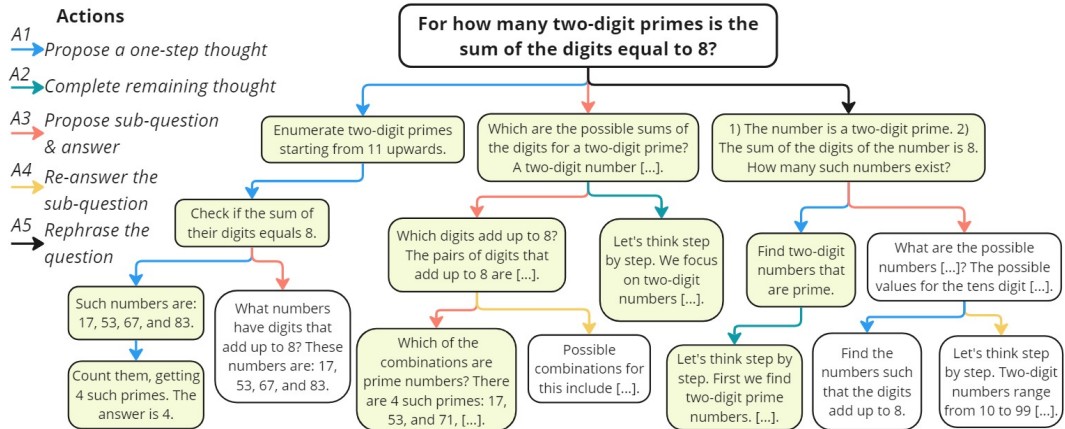

Figure 3: An example to illustrate the process of self-generator. Highlighted nodes from top to bottom constitute a complete reasoning trace. Given a question, MCTS augments the target SLM to explore a rich, human-like reasoning action space and generate the next steps based on the current state.

self-rewarding (Appendix A.1). Training a reward model, as in M* (Kang et al., 2024), can address this challenge but faces difficulties in collecting training data and generalizing across various tasks.

**Overview**. To address these challenges, this section introduces our methodology, rStar, which decomposes reasoning into solution generation and mutual verification in Fig. 2. To tackle the first challenge, we introduce a richer set of human-like reasoning actions that allows for thorough space exploration across diverse reasoning tasks. To address the second challenge, we design an SLM-tailored reward function to evaluate intermediate steps, avoiding reliance on their often unreliable self-evaluations. Moreover, we use another SLM as a discriminator to augment the MCTS process, mutually verifying the correctness of each trajectory with the generator SLM.

## 3.2 SELF-GENERATING REASONING TRAJECTORY WITH MCTS ROLLOUT

**A Rich Set of Human-like Reasoning Actions**. At the core of MCTS generation lies the action space, which defines the scope of tree exploration. Most MCTS-based methods use a single action type to build the tree. For instance, in RAP, the action is to propose the next sub-question, whereas in AlphaMath (Chen et al., 2024a) and M* (Kang et al., 2024), the action is to generate the next reasoning step. However, relying on a single action type can easily lead to ineffective space exploration.

To address this, we revisit how humans approach reasoning. Different people solve problems using diverse actions: some break into sub-questions, others solve it directly, and some might rephrase the problem to focus on key conditions. Moreover, people adjust their approach based on current states, choosing different actions as needed. Inspired by this human reasoning process, we introduce a richer set of 5 actions to maximize the SLM's potential for correctly solving complex reasoning problems.

⋄ **A1: Propose an one-step thought**. This action prompts the LLM to generate the next one-step thought for a given question, by considering the existing reasoning steps. Unlike the CoT, which generates complete thoughts, this approach simplifies the reasoning process and allows the LLM to perform better decision making (Yao et al., 2024; Besta et al., 2024).

⋄ **A2: Propose the remaining thought steps**. Instead of generating only one step thought per state, this action aligns with standard CoT, enabling "fast thinking" to solve simple question in fewer steps. Given the already generated reasoning steps, it prompts the LLM to directly produce the remaining steps until reaching the final answer.

⋄ **A3: Propose next sub-question along with its answer**. This action is inspired by *least-to-most prompting* (Zhou et al., 2022), which breaks down a complex problem into a series of simpler sub-questions and solves them sequentially. Following RAP's implementation, we prompt the LLM to ask and then answer the next sub-question.

⋄ **A4: Answer the sub-question again**. Considering that a sub-question might not be answered correctly by **A3**, we propose this action to re-answer it. To improve accuracy, this action prompts

---

**Question**: If a+b=8, b+c=-3, and a+c= -5, what is the value of the product abc?

**Candidate Solution by SLM₁**: Let's think step by step. From the three given equations, we can solve for a, b and c. Adding all three equations yields 2a + 2b + 2c = 0, or a + b + c = 0. Substituting a + c = -5 into this equation, we have -5 + b = 0, which implies b = 5. Similarly, substituting a + b = 8 gives a = 3, and using b + c = -3 results in c = -8. Finally, the product abc = 3 x 5 x (-8) = -120. The answer is: **-120**.

**Masked Solution for SLM₂**: Let's think step by step. From the three given equations, we can solve for a, b and c. Adding all three equations yields 2a + 2b + 2c = 0, or a + b + c = 0. Substituting a + c = -5 into this equation, we have -5 + b = 0, which implies b = 5. Similarly, substituting a + b = 8 gives a = 3, and using b + c = -3 results in c = -8. Finally, the product abc = 3 x 5 x (-8) = -120. The answer is: -120.

**Completed Solution 1 by SLM₂**: Let's think step by step. From the three given equations, we can solve for a, b and c. Adding all three equations yields 2a + 2b + 2c = 0, or a + b + c = 0. We can now use this result to find each variable individually. Using the equation a + b = 8, we substitute b + c = -3 to get a + (-3 - b) = 8 - 3, leading to b = 5. With b = 5, substituting into a + b = 8, we find a = 3. Then, using b = 5 in b + c = -3, we find c = -8. Finally, the product abc = 3 × 5 × (-8) = -120. The answer is: **-120**. **(consistent)**

**Completed Solution 2 by SLM₂**: Let's think step by step. From the three given equations, we can solve for a, b, and c. Adding all three equations yields 2a + 2b + 2c = 0, or a + b + c = 0. With a + c = -5, we have -5 + b = 0, so b = 5. Now, substituting b + c = -3, we deduce that c = -3 - 5 = -2. Finally, using a + b = 8, we find a = 8 - 5 = 3. The product abc = 3 × 5 × (-2) = -30. The answer is: **-30**. **(inconsistent)**

Figure 4: The prompt example for mutual reasoning consistency.

the LLM to use few-shot CoT. Note that the original answer generated by *A3* did not use a CoT-like prompt but instead followed the least-to-most problem decomposition prompt (Zhou et al., 2022).

$\diamond$*A5: Rephrase the question.* When analyzing incorrect cases, we found that many of them are due the LLM misunderstanding the question. For example, it might miss a specific condition provided in the question. Therefore, we propose a new action to rephrase the question more simply. Specifically, we prompt the LLM to clearly list all conditions given in the problem statement.

The above 5 actions define a highly diverse action space $\{A_1, A_2, A_3, A_4, A_5\}$. At each step $i$, MCTS selects an action $a_i$ from this space. We then use this action $a_i$ to prompt LLM to generate the next reasoning step $s_i$, based on the current state, which is the previous generated trajectory $x \oplus s_1 \oplus s_2 \oplus ... \oplus s_{i-1}$. To avoid inefficient explorations and unnecessary inference costs, we introduce two simple order constraints for the different action types. Specifically, *A4* can only happen after *A3*, and *A5* can only happen after the root question. As shown in Table 1, each action plays a crucial role in improving the final reasoning accuracy.

Table 1: Ablation study on the effectiveness of our rich action space: we evaluate LLaMA3-8B on 200 sampled GSM8K questions.

| Action Space | Accuracy |
|---|---|
| $A_3$ (i.e., RAP) | 70.5 |
| $A_3 + A_5$ | 72.5 |
| $A_3 + A_4 + A_5$ | 73.5 |
| $A_2 + A_3 + A_4 + A_5$ | 74.0 |
| All ($A_1 + A_2 + A_3 + A_4 + A_5$) | **75.0** |

**Reward Function.** Another critical component in MCTS is the reward function, which evaluates the value of each action and directs the tree expansion. We design a simple yet effective reward function for SLMs. First, we exclude self-rewarding techniques for any intermediate nodes due to the limited capabilities of SLMs. Second, to ensure generalization across different reasoning tasks, we avoid introducing external supervision (e.g., tools or trained value models). Our approach draws inspiration from AlphaGo (Silver et al., 2017), where we score each intermediate node based on its contribution to the final correct answer. Consequently, actions that frequently lead to correct answers receive higher rewards, making them more likely to be selected in future MCTS tree expansions.

We define $Q(s, a)$ as the reward value for node $s$ generated under action $a$. Initially, all unexplored nodes are assigned $Q(s_i, a_i) = 0$, leading to random tree expansions. Upon reaching the first terminal node $n_d$, we compute a reward score $Q(s_d, a_d)$ based on whether it reaches the correct answer. This score is then back-propagated to each intermediate node along the trajectory $\mathbf{t} = x \oplus s_1 \oplus s_2 \oplus ... \oplus s_d$. Specifically, for each $s_i$ (for $i = 1, 2, ..., d-1$), its $Q$ value is updated as follows: $Q(s_i, a_i) = Q(s_i, a_i) + Q(s_d, a_d)$. To compute the $Q(s_d, a_d)$ for the terminal node, we use the likelihood (confidence) of self-consistency majority voting as the reward value.

**Solution Generation with MCTS Rollout**. We now describe how our MCTS generates candidate reasoning trajectories. Starting from the initial root node $s_0$, we perform multiple searches consisting of *selection*, *expansion*, *simulations* and *back-propagation*. Specifically, the simulation is performed using the default *rollout* policy, and to achieve more accurate reward estimation, we perform multiple rollouts. To balance the exploration and exploitation, we use the well-known Upper Confidence

Bounds applied to Trees (UCT) (Kocsis & Szepesvári, 2006) to select each node. This selection process is mathematically represented as:

$$\text{UCT}(s, a) = \frac{Q(s, a)}{N(s, a)} + c \sqrt{\frac{\ln N_{parent}(s)}{N(s, a)}}.$$

where $N(s, a)$ is the number of times node $s$ has been visited in previous iterations, and $N_{parent}(s)$ represents the visiting count of the parent node of $s$. $Q(s, a)$ is the estimated reward value and will be updated through back-propagation. $c$ is a constant that balances exploitation and exploration.

Once the search reaches a terminal node, either a terminal state or a predetermined maximum tree depth $d$, we obtain a trajectory from the root to terminal node. We collect all trajectories from the rollout iterations as candidate solutions. The next section explains how we verify each of them.

### 3.3 REASONING TRAJECTORY SELECTION WITH MUTUAL CONSISTENCY

In traditional MCTS, typically only one trajectory is selected as the final solution based on a specific metric, such as choosing the path with the highest reward from the rollout iterations. Unfortunately, after trying various existing methods, we found it challenging to define a single metric that reliably selects the trajectory containing the correct answer. Therefore, we collect all trajectories and propose mutual reasoning consistency for answer selection.

**Mutual Reasoning Consistency by Discriminator SLM$_2$.** As shown in Fig. 2, in addition to the target SLM $M$, we introduce another SLM $\hat{M}$ to serve as a discriminator, providing external unsupervised feedback for each candidate trajectory. Specifically, for $\mathbf{t} = x \oplus s_1 \oplus s_2 \oplus ... \oplus s_d$, we mask the reasoning steps starting from a randomly sampled step $i$ ($i < d$). We then provide the earlier reasoning trajectory $\mathbf{t} = x \oplus s_1 \oplus s_2 \oplus ... \oplus s_{i-1}$ as a prompt to $\hat{M}$ to complete the remaining steps for the question. Due to the provision of the earlier $i - 1$ reasoning steps as a hint, we reduce the difficulty, thereby increasing the likelihood that SLM $\hat{M}$ can provide the correct answer. As shown in Fig. 4, we compare whether the answer completed by $\hat{M}$ matches the original trajectory $\mathbf{t}$. If they are consistent, we consider $t$ as an validate trajectory for final selection.

We provide an intuitive explanation to illustrate the rational behind our approach. Consider students solving a problem without a teacher's feedback. A student (SLM$_1$) unsure of their solution might ask a peer (SLM$_2$) to review their reasoning. If the peer, given the same initial steps, arrives at the same answer, the student gains confidence in their solution. This peer verification process reflects the mutual reasoning consistency we aim to achieve.

**Final Trajectory Selection by SLM$_1$.** After applying mutual reasoning consistency to all candidate trajectories, we return to the target SLM $M$ to select the final trajectory from the validated ones. We compute each trajectory's final score by multiplying its reward with the terminal node's confidence score achieved from rollouts. The trajectory with the highest final score is chosen as the solution.

## 4 EXPERIMENTS

### 4.1 SETUP

**Models and Datasets.** rStar is a general approach applicable to various LLMs and reasoning tasks. We evaluate 5 SLMs: Phi3-mini (3.8B) (Abdin et al., 2024), LLaMA2-7B, Mistral-7B (Jiang et al., 2023), LLaMA3-8B, and LLaMA3-8B-Instruct (Meta, 2024). We test across 5 reasoning tasks, including 4 mathematical tasks (GSM8K (Cobbe et al., 2021), GSM-Hard (Gao et al., 2022), MATH (Hendrycks et al., 2021), SVAMP (Patel et al., 2021)) and one commonsense reasoning task (StrategyQA (Geva et al., 2021)).

**Implementation Details.** In the trajectory self-generation stage, we augment each target SLM with our MCTS, performing 32 rollouts. Except for MATH, where we set the depth $d$ to 8, all other tasks have a $d$=5. Actions $A_1$ and $A_3$ have a maximum of 5 nodes per depth, while the other actions have a default node count of 1. In the trajectory discrimination stage, we use Phi3-mini-4k as the discriminator, which has only 3.8B parameters, for effective inference. Moreover, the discriminator performs inference in a parallelized manner, making the verification process highly efficient. Notably, when Phi3 is the target SLM, it performs self-discrimination. For a given trajectory, we randomly split it between 20% and 80% of its steps, providing the first half of the steps as input to the discriminator SLM, which then completes the remaining steps. Detailed prompts are available in appendix A.5.

Table 2: rStar greatly improves reasoning accuracy across various SLMs and tasks. rStar (generator@maj): uses majority voting for answer verification to show the MCTS generator's effectiveness.

| Method | LLaMA2-7B | Mistral-7B | LLaMA3-8B | LLaMA3-8B-Instruct | Phi3-mini-4k |
|---|---|---|---|---|---|
| *GSM8K* | | | | | |
| Zero-shot CoT | 1.44 | 17.89 | 22.66 | 68.38 | 20.17 |
| Few-shot CoT | 12.51 | 36.46 | 47.23 | 74.53 | 83.45 |
| SC@maj8 | 15.31 | 42.91 | 54.21 | 78.39 | 86.35 |
| SC@maj64 | 20.77 | 52.84 | 64.37 | 83.24 | 88.02 |
| SC@maj128 | 23.05 | 57.25 | 67.55 | 84.69 | 88.68 |
| ToT | 12.96 | 38.89 | 36.01 | 69.07 | 79.68 |
| RAP | 24.34 | 56.25 | 57.99 | 80.59 | 81.88 |
| **rStar (generator @maj)** | **27.22** | **64.59** | **74.38** | **88.70** | **90.44** |
| **rStar** | **63.91** | **81.88** | **85.52** | **91.13** | **90.67** |
| *GSM-Hard* | | | | | |
| Zero-shot CoT | 0.83 | 5.16 | 6.44 | 14.94 | 33.73 |
| Few-shot CoT | 3.71 | 13.57 | 13.80 | 25.63 | 40.63 |
| SC@maj8 | 4.39 | 17.36 | 18.20 | 28.51 | 42.00 |
| SC@maj64 | 6.52 | 22.59 | 23.73 | 30.33 | 44.80 |
| SC@maj128 | 6.89 | 25.01 | 25.47 | 31.16 | 45.56 |
| ToT | 2.35 | 11.47 | 10.61 | 19.64 | 32.68 |
| RAP | 7.28 | 22.52 | 18.95 | 29.64 | 40.94 |
| **rStar (generator @maj)** | **8.64** | **29.26** | **26.76** | **33.35** | **46.55** |
| **rStar** | **18.57** | **37.91** | **32.97** | **37.53** | **46.55** |
| *SVAMP* | | | | | |
| Zero-shot CoT | 8.90 | 26.10 | 40.20 | 70.90 | 84.70 |
| Few-shot CoT | 48.10 | 72.80 | 76.90 | 89.20 | 92.80 |
| SC@maj8 | 49.90 | 74.60 | 79.10 | 89.20 | 93.50 |
| SC@maj64 | 54.10 | 76.70 | 80.70 | 90.50 | 93.30 |
| SC@maj128 | 54.50 | 76.60 | 80.80 | 90.60 | 93.70 |
| ToT | 33.40 | 56.30 | 62.20 | 79.80 | 84.90 |
| RAP | 41.00 | 71.80 | 73.10 | 85.70 | 91.50 |
| **rStar (generator @maj)** | **60.30** | **83.10** | **86.20** | **91.89** | **93.80** |
| **rStar** | **74.90** | **86.40** | **90.00** | **94.29** | **94.10** |
| *StrategyQA* | | | | | |
| Zero-shot CoT | 52.67 | 57.20 | 41.48 | 57.21 | 54.68 |
| Few-shot CoT | 58.82 | 65.65 | 64.05 | 68.41 | 63.61 |
| SC@maj8 | 59.10 | 65.50 | 63.76 | 68.26 | 64.34 |
| SC@maj64 | 58.51 | 63.61 | 63.46 | 67.39 | 62.74 |
| SC@maj128 | 58.37 | 62.01 | 63.31 | 66.67 | 59.53 |
| ToT | 45.27 | 55.75 | 57.64 | 60.41 | 40.47 |
| RAP | 59.68 | 64.48 | 63.32 | 68.71 | 60.26 |
| **rStar (generator @maj)** | **61.57** | **69.43** | **65.50** | **71.47** | **65.50** |
| **rStar** | **67.25** | **70.31** | **67.69** | **71.57** | **67.25** |

Table 3: Reasoning performance comparison on the challenging MATH-500 dataset. Due to the extensive LaTeX syntax in the dataset, which is challenging for pre-trained LLMs in instruction following, we evaluate only on LLaMA3-8B-instruct and Phi3-Mini-4k-Instruct.

| Method | LLaMA3-8b-Instruct | Phi3-mini-4k |
|---|---|---|
| Zeroshot CoT | 5.80 | 3.60 |
| Fewshot CoT | 23.40 | 32.20 |
| SC@maj8 | 30.00 | 40.40 |
| SC@maj64 | 33.00 | 45.20 |
| SC@maj128 | 33.80 | 45.60 |
| ToT | 13.60 | 18.20 |
| RAP | 18.80 | 27.80 |
| **rStar (generator @maj)** | **38.30** | **48.40** |
| **rStar** | **42.94** | **48.60** |

## 4.2 MAIN RESULTS

**Baselines**. We compare rStar against three strong baseline types: **(i)** *single-round CoT prompting*, including zero-shot CoT (Kojima et al., 2022) and few-shot CoT (Wei et al., 2022); **(ii)** *multi-round CoT prompting* using the widely adopted self-consistency (SC) method (Wang et al., 2023). We sample answers 8, 64, and 128 times, employing majority voting for answer selection; and **(iii)** *multi-round self-improving approaches*. For this, we select ToT (Yao et al., 2024) and RAP (Hao et al., 2023) as baselines, using BFS and MCTS for tree search, respectively. Note that the action in ToT corresponds to our action $A_1$, while RAP corresponds to our action $A_3$. For the answer selection, we follow their original implementations.

**Results on diverse reasoning benchmarks**. We start by evaluating the effectiveness of rStar on general reasoning benchmarks. Table 2 compares its accuracy with state-of-the-art baselines on diverse SLMs and reasoning datasets. To demonstrate the effectiveness of our generator, we also provide the accuracy of rStar (gen. @maj), which do not apply our discriminator and use majority voting for answer verification. We highlight three key observations: **(1)** SLMs empowered with rStar demonstrate highly capable problem-solving abilities. For example, LLaMA2-7B originally had an accuracy of only 12.51% on GSM8K using few-shot CoT. Surprisingly, with improvements from rStar, its accuracy increased to 63.91%, nearly matching the accuracy achieved with fine-tuning as shown in Fig. 1. Similarly, Mistral with rStar can even outperform fine-tuned MetaMath by +4.18%. This improvement shows that SLMs already have strong reasoning capabilities but need guidance to generate and select the correct solutions. **(2)** rStar consistently improves the reasoning accuracy of various evaluated SLMs across different tasks to a state-of-the-art level. In contrast, none of the baseline approaches consistently perform well across all four benchmarks. For example, while SC excels in three mathematical tasks, it is less effective on the logical reasoning task of StrategyQA. Specifically, SC with more sampling can even lower the score on StrategyQA. RAP performs better than SC on StrategyQA but falls short compared to SC on most mathematical reasoning tasks. **(3)** Even without our proposed discriminator for reasoning trajectory verification, our MCTS generator demonstrates greater effectiveness in improving reasoning accuracy for SLMs compared to existing multi-round inference baselines. For example, rStar (generator @maj) achieves up to 2.88%-16.39% higher accuracy than RAP, 10.60%- 38.37% higher accuracy than ToT, and 1.69% - 7.34% higher accuracy than SC on the GSM8K dataset.

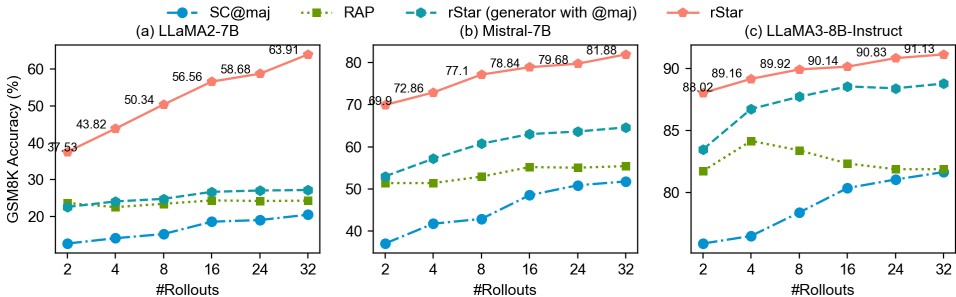

Figure 5: Performance comparison on the GSM8K dataset under different number of rollouts. rStar can significantly improve reasoning accuracy with just 2 rollouts.

Table 4: Ablation study on the effectiveness of our MCTS generator. Ours+self-eval: we apply self-evaluation to prompt model for self-rewarding each intermediate action in our generator.

| Generator | LLaMA3-8B | | | | LLaMA3-8B-Instruct | | | |
|---|---|---|---|---|---|---|---|---|
| | GSM8K Answer verification | | StrategyQA Answer verification | | GSM8K Answer verification | | StrategyQA Answer verification | |
| | Maj | Ours | Maj | Ours | Maj | Ours | Maj | Ours |
| RAP | 56.56 | 57.31 | 62.30 | 64.63 | 81.35 | 84.69 | 69.43 | 70.60 |
| SC (@128) | 67.55 | 85.06 | 63.31 | 65.65 | 84.69 | 89.99 | 66.67 | 68.56 |
| Ours+Self-eval | 70.28 | 82.18 | 65.07 | 66.22 | 88.07 | 89.92 | 69.28 | 69.43 |
| **Ours** | **74.38** | **85.52** | **65.50** | **67.69** | **88.70** | **91.13** | **71.47** | **71.57** |

Table 5: Ablation study on discriminator effectiveness. We evaluate accuracy on GSM8K. *Left*: Our discriminator consistently outperforms others in verifying solution trajectories generated by different generators. *Right*: The ablation study on the choice of discriminator model.

| Discriminator | LLaMA3-8B Generator | | LLaMA3-8B-Instruct Generator | |
|---|---|---|---|---|
| | SC | Ours | SC | Ours |
| Maj | 67.55 | 74.38 | 84.69 | 88.70 |
| Self-verification | 74.00 | 75.52 | 83.02 | 86.63 |
| **Ours** | **85.06** | **85.52** | **89.99** | **91.13** |

| Model | Discriminator SLM | Accuracy |
|---|---|---|
| | Maj | 88.70 |
| | LLaMA3-8B-Instruct | **88.78** |
| LLaMA3-8B-Instruct | LLaMA3.1-8B-Instruct | **89.52** |
| | Phi3-Mini-Instruct | **91.13** |
| | GPT-4 (2024-05-01) | **92.57** |

**Results on challenging mathematical dataset**. We also evaluate the effectiveness of rStar on more challenging mathematical datasets. In particular, we select the GSM-Hard and MATH datasets. Following (Wang et al., 2024b; Lightman et al., 2023), we use MATH-500, a subset of representative problems from the MATH dataset, to speedup the evaluation. As shown in Table 2 and Table 3, rStar is capable of significantly improve the reasoning accuracy of SLMs on these challenging mathematical datasets. Remarkably, when compared to SOTA baselines, we observe a significant improvements of up to +12.9% and +9.14% on GSM-Hard and MATH-500, respectively.

## 4.3 ABLATION STUDY

**Effectiveness under different rollouts**. rStar uses a rollout policy for MCTS tree expansion. More rollouts generate more candidate solution trajectories but increase inference cost. In Fig. 5, we compare the accuracy of SC, RAP, and our rStar across different rollouts on GSM8K. For SC, we sample solutions based on each number of rollouts and use majority voting to select the answer. We highlight two key observations: **(1)** Even with just 2 rollouts, rStar significantly improves reasoning accuracy for SLMs, demonstrating its effectiveness; **(2)** Both rStar and SC benefit from more rollouts, whereas RAP tends to saturate and even decline after 4 rollouts on LLaMA3-8B-Instruct. One reason is that the single-type action space in RAP limits the effective MCTS exploration.

**The effectiveness of MCTS generator**. We compare our MCTS generator with three baselines: (i) the MCTS generator used in RAP; (ii) SC with 128 randomly sampled solutions; and (iii) our generator with Self-evaluation, a popular technique that self-evaluates the reward score for each action. Baseline (iii) specifically evaluates the effectiveness of our reward function. To isolate the impact of answer verification methods, each generator is evaluated under both majority voting and our discriminator for trajectory selection. As shown in Table 4, our generator consistently outperforms the baseline generators across different answer verification methods. More, we demonstrate the effectiveness of our SLM-tailored reward function, as self-evaluation reduces our generator's accuracy.

**The effectiveness of discriminator**. We setup two experiments for evaluation. First, we compare our discrimination approach with two baselines: the majority voting and self-verification (Weng et al., 2023). Specifically, we follow the key idea in Weng et al. (2023) to prompt the SLM (i.e., generator SLM) to self-verify the correctness of each trajectory. To demonstrate the generalization ability of our discriminator, we used candidate solutions from different generators for evaluation. As shown in Table 5 (Left), our discriminator significantly improves reasoning accuracy when performing answer verification on trajectories generated by different generators. Similar to the previous self-evaluation experiment, self-verification on SLMs is ineffective.

Second, we study the impact of discriminator model selection. Our current discriminator models are all Phi3-Mini-Instruct. We tested various LLMs, both stronger and weaker, as discriminators for LLaMA3-8B-Instruct. As shown in Table 5 (Right), the choice of discriminator model generally does

not affect the effectiveness of our mutual reasoning consistency for answer verification. Notably, using the powerful GPT-4 as the discriminator only slightly improves performance (91.13% to 92.57%), demonstrating that mutual reasoning consistency can effectively verify answers using SLMs.

## 5 CONCLUSION

In this work, we present rStar, a generator-discriminator self-play approach that significantly grow the reasoning capabilities for SLMs at the inference time. Our approach reveals that SLMs, such as LLaMA2-7B, already exhibit strong reasoning capabilities prior to domain specialized supervised fine-tuning. rStar achieves state-of-the-art performance across five SLMs and five diverse reasoning tasks, substantially outperforming existing multi-round prompting and self-improvement approaches. Furthermore, we conduct extensive ablation studies and analysis, contributing to the development of more advanced SLM self-improved reasoning.

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

# A APPENDIX

## A.1 EXPERIMENTS TO EVALUATE THE SELF-REWARDING IN SLMS

Table 6: Analysis on the effectiveness of SLMs' self-rewarding. The original $r_1$ is a self-evaluation of the helpfulness of the new proposed subquestion, while $r_2$ measures the confidence in answering the subquestion through self-consistency majority voting. Results show that replacing the self-evaluated $r_1$ to random values does not significantly impact the final reasoning performance.

| Method | LLaMA2-7B | Mistral |
|---|---|---|
| *GSM8K* | | |
| RAP | 24.34 | 56.25 |
| **RAP + random** $r_1$ | **22.90** | **55.50** |
| RAP + random $r_2$ | 22.67 | 49.66 |
| *Multiarith* | | |
| RAP | 57.22 | 91.11 |
| **RAP + random** $r_1$ | **52.78** | **90.56** |
| RAP + random $r_2$ | 47.22 | 81.11 |

**Ablation study on self-rewarding in RAP**. RAP rewards both intermediate and terminal nodes. For each node generated by its action, it combines two scores, $r_1$ and $r_2$, to determine the final reward score. Formally, $r = r_1 \times r_2$. $r_1$ is a self-evaluation score that evaluates the LLM's own estimation of the helpfulness of the current node. Specifically, it prompts the LLM with the question "Is the new question useful?". $r_2$ is the confidence of correctly answering the proposed new question, measured by self-consistency majority voting.

To evaluate the effectiveness of self-rewarding in RAP, we replace $r_1$ and $r_2$ with random values sampled from (0,1)and re-run RAP on LLaMA2-7B and Mistral-7B. We select a challenging dataset, GSM8K and an easy mathematical reasoning dataset, Multiarith (Roy & Roth, 2015), for evaluation.

Table 6 compares the results with original RAP. We can see that replacing $r_1$ with random values has minimal impact on RAP's performance across different SLMs and datasets. However, replacing $r_2$ with random values result in a noticeable drop in accuracy on Mistral and Multiarith. This indicates that self-evaluation $r_1$ has minimal effect, suggesting that LLaMA2-7B and Mistral are essentially performing near-random self-evaluations.

## A.2 ADDITIONAL RESULTS

Table 7: Performance on the competition mathematical AMC23 dataset.

| Method | Phi3-mini |
|---|---|
| Zeroshot CoT | 2/40 |
| Fewshot CoT | 4/40 |
| SC@maj64 | 5/40 |
| SC@maj128 | 5/40 |
| **rStar** | **9/40** |

**Results on competition math problems**. In addition to challenging datasets like GSM-Hard and MATH, we evaluate the effectiveness of rStar on the ACM Math Competition 2023 (AMC23). As shown in Table 7, Phi3-mini 3.8B, with few-shot prompting, can correctly answers only 4 questions. Self-consistency multi-rounds inference provides a slight improvement to 5. However, with rStar, Phi3-mini solves 9 questions, achieving a relative improvement of over 100%.

**SLMs with rStar on unseen challenging math benchmarks**. To show that rStar truly enhances the reasoning capabilities of SLMs, rather than just allowing them to better exploit statistical patterns from previously seen data, we test it on 22 problems from the AMC 2024, which were released in January 2024. Since the SLMs (Mistral and LLaMA3-8B) were trained on data available before December 2023, there is no data leakage on the AMC 2024. As shown in Table 8, rStar substantially improves the two SLMs performance on the challenging AMC 2024 benchmark.

Table 8: Performance on AMC 2024. rStar enables SLMs to better reason on unseen challenging math benchmarks.

| Method | Mistral-7B-v0.1 (cutoff: before Oct. 2023) | LLaMA3.1-8B-Instruct (cutoff: before Dec. 2023) |
|---|---|---|
| Few-shot CoT | 22.72% | 18.18% |
| SC@maj128 | 18.18% | 31.82% |
| **rStar** (32 rollouts) | **31.82%** | **40.91%** |

Table 9: GSM8K accuracy on larger Mistral-Nemo-Instruct (12B).

| Method | Phi3-mini |
|---|---|
| Zeroshot CoT | 75.8 |
| Fewshot CoT | 84.2 |
| SC@maj32 | 87.1 |
| SC@maj128 | 87.4 |
| **rStar** | **91.1** |

**rStar on larger LLMs**. To further demonstrate the effectiveness of rStar, we conduct experiments using a 12B LLM (Mistral-Nemo-Instruct-12B) with 32 rollouts. The results on GSM8K dataset are summarized in Table 9. We can see that rStar remains effective when scaled to larger LLMs, demonstrating its potential for broader applicability.

A.3  DISCUSSIONS

Table 10: Inference costs of rStar on GSM8K. We show the average number of inferences and generated tokens required to solve a question.

| | LLaMA2-7B | Mistral |
|---|---|---|
| Avg. calls | 166.81 | 148.90 |
| Avg. generated tokens | 367.1k | 348.6k |

**Discussions on the importance of generator and discriminator**. In our experiments, we found that on certain SLMs, the discriminator yields more significant improvement than the generator. For instance, on LLaMA2-7B, rStar (generator @maj) can improves accuracy by +4.17% on GSM8K, while our discriminator can further boosts accuracy by +36.69%. However, both the generator and discriminator are crucial to final performance. The generator must effectively produce the correct solution first, and the discriminator then provide reasonable feedback to select the correct solution.

The importance of the generator and discriminator varies based on the SLM's solution generation effectiveness. For LLaMA2-7B, the proportion of correct solution trajectories is low, leading to a lower majority voting score. In this case, the discriminator is crucial for identifying the correct solution, significantly boosting accuracy. Conversely, for stronger models like LLaMA3-8B-instruct, our generator produces a higher proportion of correct solutions. Therefore, improving the generator results in greater accuracy improvements.

**Inference cost**. rStar grows SLMs reasoning capabilities at inference time. The primary inference cost arises from our MCTS self-generator. Table 10 shows the average number of inferences and tokens generated for solving a GSM8K question after 32 rollouts. On LLaMA2-7B and Mistral, this averages 166 and 148 model calls to solve a question, respectively. Currently, completing the 32 rollouts for the entire GSM8K test set takes about 4.5 days on a single A100 GPU per model. These costs can be significantly reduced by distributing tasks across multiple GPUs or batching model calls within each rollout. Moreover, by increasing the batch size (i.e., performing MCTS rollouts for multiple problems simultaneously), we can improve GPU utilization and accelerate inference time, making rStar more inference-friendly.

**Discussions on mutual consistency**. rStar leverages SLMs as discriminators for mutual answer verification. A key advantage is the SLM's strong generalization compared to specialized reward models, eliminating the need for costly training data. Our mutual consistency can complement existing reward models, as multiple reward signals from both SLM and reward models improves

solution verification. We also observed that, at times, the partial trajectory fed to the discriminator SLM contains errors. In such cases, using the partial trace as a hint for the same generator SLM may lead to repeated mistakes. However, due to differences in training data and algorithms, the discriminator SLM typically produces a different incorrect answer, which causes a mismatch with the generator SLM and is subsequently filtered out. In rare instances, if an early mistake overly simplifies the problem, the discriminator SLM may replicate the error.

## A.4 EXAMPLES OF REASONING TRAJECTORY

---

**Example 1: rStar vs. Few-shot Prompting**

Question: Trent is 5 years older than Jane, and Jane is 3 years younger than Quinn. If Quinn is 30, how old is Trent?

**LLaMA2-7B's wrong response under Few-shot CoT prompting**:
Let's think step by step. Trent is 5 years older than Jane. Jane is 3 years younger than Quinn. So Trent is 5 + 3 = 8 years older than Quinn. 30 - 8 is 22. The answer is: 22.

**LLaMA2-7B's correct response under rStar**:
Node1: [$A_3$] How old is Jane? subanswer: Jane is 3 years younger than Quinn, who is 30 years old. This means she is 30 - 3 = 27 years old. The answer is 27.
Node2: [$A_3$] How old is Trent? subanswer: Trent is 5 years older than Jane, who is 27 years old. This means he is 27 + 5 = 32 years old. The answer is 32.
Node3: [$A_2$] Let's think step by step. Trent is 5 years older than Jane, who is 27 years old. That makes Trent 27 + 5 = 32 years old. The answer is: 32.

---

**Example 2: rStar under different action spaces**

Question: Mike needed a new pair of jeans. When he got to the mall he saw that his favorite jeans were advertised 25% off. The original price of the jeans was $40. How much money will Mike have left over if he pays with a $50.00 bill?

**LLaMA2-7B's incorrect response using rStar with only action $A_1$**:
Node1: [$A_1$] Mike needs a pair of jeans, but there is a discount so it costs 25% less than the original price of $40.
Node2: [$A_1$] Since we know the original price, we can use this information and say that the discount is ($40 ŏ0d7 0.25) = 10 dollars.
Node3: [$A_1$] The jeans cost 10 dollars less since the sale is 25%. So 50 - 10 is 40. This is the amount Mike spent.
Node4: [$A_1$] The answer is: 40.

**LLaMA2-7B's incorrect response using rStar with only action $A_3$**:
Node1: [$A_3$] What is the discounted price of the jeans? subanswer: The original price of the jeans is $40. The discount is 25%, so the discounted price is (1 - 0.25) * $40 = $30. The answer is 30.
Node2: [$A_3$] How much money will Mike have left over if he pays with a $50.00 bill?
subanswer: If he pays with a $50 bill, he will have 30 - 50 = -$20 left over. The answer is -20.

**LLaMA2-7B's correct response 1 under rStar**:
Node1: [$A_1$] The original price is $40.
Node2: [$A_1$] The discount amount is 25%.
Node3: [$A_2$]: Let's think step by step. Mike needed a new pair of jeans. His favorite jeans were originally priced at $40. When 25% off was applied, it became $30. Mike paid with a $50 dollar bill, so he had $20 extra (because $50 - $30 = $20). The answer is: 20.

**LLaMA2-7B's correct response 2 under rStar**:
Node1: [$A_3$]: How much money does Mike need to pay for a pair of jeans? subanswer: With a discount of 25%, the jeans cost $40 - (40 * 0.25) = $30. The answer is $30.
Node2: [$A_2$]: Let's think step by step. Mike needs 30 dollars. Paying it with a 50 dollar bill leaves him with 20 dollars extra. The answer is: 20.

---

**Example 3: Mutual reasoning consistency**

Question: Josh decides to try flipping a house. He buys a house for $80,000 and then puts in $50,000 in repairs. This increased the value of the house by 150%. How much profit did he make?

**Error trace generated by LLaMA3-8B-instruct**:
Let's think step by step. The new value of the house is $80,000 + (150% * $80,000) = $80,000 + $120,000 = $200,000 (wrong step,note that we feed this step along with the preceding steps as a partial trace to the discriminator SLM). Josh bought the house for $80,000 and sold it for $200,000. His profit is $200,000 - $80,000 = $120,000. Now we can answer the question: Josh made a profit of $120,000. The answer is 120,000.

**Correct solution 1 generated by discriminator SLM (phi3-mini-instruct)**:
But don't forget about the cost of repairs! To get his actual net profit, we subtract $50,000 (the repair costs) from $120,000 (profit). Therefore, Josh made a profit of $120,000 - $50,000 = $70,000. The answer is: 70000.

**Correct solution 2 generated by discriminator SLM (phi3-mini-instruct)**:
Also consider the cost of $50,000 repairs which makes his total investment $130,000 ($80,000 + $50,000). To calculate his net profit, we subtract his total investment from his selling price, $200,000 - $130,000 = $70,000. The answer is: 70000.

**Wrong solution generated by discriminator SLM (phi3-mini-instruct)**:
However, we also need to consider the cost of repairs which totals to $50,000. Therefore, to find out the actual profit made, we subtract this repair cost from the initial purchase price plus profits i.e. ($80,000 + $120,000) - $50,000, resulting in a profit of $150,000. So, the answer is: 150000.

A.5   PROMPT EXAMPLES IN MTCS

---

**Action 1. Propose an one-step thought.**

### Instruction: There are 15 trees in the grove. Grove workers will plant trees in the grove today. After they are done, there will be 21 trees. How many trees did the grove workers plant today?

### Response: Let's think step by step.
Step 1: Identify the initial number of trees. The problem states there are 15 trees in the grove.
Step 2: Identify the final number of trees. The problem states there will be 21 trees after the workers are done planting.
Step 3: Subtract the initial number of trees from the final number of trees to find out how many trees were planted.
Step 4: Therefore, the grove workers planted 21 (final number of trees) - 15 (initial number of trees) = 6 trees today.
Step 5: The answer is 6.

### Instruction: If there are 3 cars in the parking lot and 2 more cars arrive, how many cars are in the parking lot?

### Response: Let's think step by step.
Step 1: Start with the number of cars that are already in the parking lot, which is 3 cars.
Step 2: Add the number of cars that arrive, which is 2 cars.
Step 3: Add the numbers together. there are 3 cars + 2 cars = 5 cars in the parking lot. Step 4: The answer is 5.

### Instruction: Leah had 32 chocolates and her sister had 42. If they ate 35, how many pieces do they have left in total?

### Response: Let's think step by step.
Step 1: Add the number of chocolates Leah and her sister had initially. Leah had 32 chocolates and her sister had 42 chocolates. So, they had 32 + 42 = 74 chocolates in total.
Step 2: Subtract the number of chocolates they ate from the total number of chocolates they had. They ate 35 chocolates. So, they have 74 - 35 = 39 chocolates left.
Step 3: The answer is 39.

### Instruction: Jason had 20 lollipops. He gave Denny some lollipops. Now Jason has 12 lollipops. How many lollipops did Jason give to Denny?

### Response: Let's think step by step.
Step 1: Identify the initial number of lollipops. The problem states Jason had 20 lollipops.
Step 2: Identify the final number of lollipops. The problem states Jason now has 12 lollipops.
Step 3: Subtract the final number of lollipops from the initial number to find out how many lollipops Jason gave to Denny.
Step 4: Therefore, Jason gave Denny 20 (initial number of lollipops) - 12 (final number of lollipops) = 8 lollipops.
Step 5: The answer is 8.

### **Instruction: {user question}**
### **Response: Let's think step by step.**

---

---

**Action 2: Propose the remaining thought steps /A4: Answer the sub-question again.**

---

### Instruction: There are 15 trees in the grove. Grove workers will plant trees in the grove today. After they are done, there will be 21 trees. How many trees did the grove workers plant today?

### Response: Let's think step by step. There are 15 trees originally. Then there were 21 trees after some more were planted. So there must have been 21 - 15 = 6. The answer is: 6.

### Instruction: If there are 3 cars in the parking lot and 2 more cars arrive, how many cars are in the parking lot?

### Response: Let's think step by step. There are originally 3 cars. 2 more cars arrive. 3 + 2 = 5. The answer is: 5.

### Instruction: Leah had 32 chocolates and her sister had 42. If they ate 35, how many pieces do they have left in total?

### Response: Let's think step by step. Originally, Leah had 32 chocolates. Her sister had 42. So in total they had 32 + 42 = 74. After eating 35, they had 74 - 35 = 39. The answer is: 39.

### Instruction: Jason had 20 lollipops. He gave Denny some lollipops. Now Jason has 12 lollipops. How many lollipops did Jason give to Denny?

### Response: Let's think step by step. Jason started with 20 lollipops. Then he had 12 after giving some to Denny. So he gave Denny 20 - 12 = 8. The answer is: 8.

### Instruction: Shawn has five toys. For Christmas, he got two toys each from his mom and dad. How many toys does he have now?

### Response: Let's think step by step. Shawn started with 5 toys. If he got 2 toys each from his mom and dad, then that is 4 more toys. 5 + 4 = 9. The answer is: 9.

### Instruction: There were nine computers in the server room. Five more computers were installed each day, from monday to thursday. How many computers are now in the server room?

### Response: Let's think step by step. There were originally 9 computers. For each of 4 days, 5 more computers were added. So 5 * 4 = 20 computers were added. 9 + 20 is 29. The answer is: 29.

### Instruction: Michael had 58 golf balls. On tuesday, he lost 23 golf balls. On wednesday, he lost 2 more. How many golf balls did he have at the end of wednesday?

### Response: Let's think step by step. Michael started with 58 golf balls. After losing 23 on tuesday, he had 58 - 23 = 35. After losing 2 more, he had 35 - 2 = 33 golf balls. The answer is: 33.

### Instruction: Olivia has $23. She bought five bagels for $3 each. How much money does she have left?

### Response: Let's think step by step. Olivia had 23 dollars. 5 bagels for 3 dollars each will be 5 x 3 = 15 dollars. So she has 23 - 15 dollars left. 23 - 15 is 8. The answer is: 8.

### **Instruction: {user question}**
### **Response**:

---

**Action 3: Propose next sub-question along with its answer.**

Given a question, please decompose it into sub-questions. For each sub-question, please answer it in a complete sentence, ending with "The answer is <a numeric answer>". When the original question is answerable, please start the subquestion with "Now we can answer the question: <original question>".

Question 1: Four years ago, Kody was only half as old as Mohamed. If Mohamed is currently twice as 30 years old, how old is Kody?
Question 1.1: How old is Mohamed currently?
Answer 1.1: Mohamed is twice as old as 30 years, which means he is 30 * 2 = 60 years old. Question 1.2: What was Kody's age four years ago, given that it was half of Mohamed's age at that time?
Answer 1.2: Four years ago, Mohamed was 60 - 4 = 56 years old, so Kody was half of that, which is 56 / 2 = 28 years old.
Question 1.3: Now we can answer the question: How old is Kody?
Answer 1.3: Kody is currently 28 + 4 = 32 years old. The answer is 32.

Question 2: On a moonless night, three fireflies danced in the evening breeze. They were joined by four less than a dozen more fireflies before two of the fireflies flew away. How many fireflies remained?
Question 2.1: How many fireflies joined?
Answer 2.1: The fireflies were joined by four less than a dozen more fireflies, which are 12 - 4 = 8 fireflies. The answer is 8.
Question 2.2: Now we can answer the question: How many fireflies remained?
Answer 2.2: Three fireflies were dancing originally. They were joined by 8 fireflies before two of them flew away. So there were 3 + 8 - 2 = 9 remaining. The answer is 9.

Question 3: Ali has four $10 bills and six $20 bills that he saved after working for Mr. James on his farm. Ali gives her sister half of the total money he has and uses 3/5 of the remaining amount of money to buy dinner. Calculate the amount of money he has after buying the dinner.
Question 3.1: How much money does Ali have after giving half of his total money to his sister?
Answer 3.1: Ali initially has four $10 bills and six $20 bills, totaling 4 * 10 + 6 * 20 = 160 dollars. Giving half of this to his sister leaves him with 160 / 2 = 80 dollars. The answer is 80.
Question 3.2: How much money does Ali spend on dinner?
Answer 3.2: Ali uses 3/5 of his remaining money, which is 80 dollars, to buy dinner. Therefore, he spends 80 * 3/5 = 48 dollars on dinner. The answer is 48.
Question 3.3: Now we can answer the question: How much money does Ali have after buying the dinner?
Answer 3.3: After buying the dinner, Ali has 80 - 48 = 32 dollars left. The answer is 32.

Question 4: A car is driving through a tunnel with many turns. After a while, the car must travel through a ring that requires a total of 4 right-hand turns. After the 1st turn, it travels 5 meters. After the 2nd turn, it travels 8 meters. After the 3rd turn, it travels a little further and at the 4th turn, it immediately exits the tunnel. If the car has driven a total of 23 meters around the ring, how far did it have to travel after the 3rd turn?
Question 4.1: How far did the car travel except for the 3rd turn?
Answer 4.1: It travels 5 meters after the 1st, 8 meters after the 2nd, and 0 meters after the 4th turn. It's a total of 5 + 8 + 0 = 13 meters. The answer is 13.
Question 4.2: Now we can answer the question: How far did the car have to travel after the 3rd turn?
Answer 4.2: The car has driven a total of 23 meters around the ring. It travels 13 meters except for the 3rd turn. So it has to travel 23 - 13 = 10 meters after the 3rd turn. The answer is 10.

**Question 5: {user question}**

---

**Action 5: Rephrase the question/sub-question.**

You are an AI assistant to help me rephrase questions by splitting the question context into conditions. In your rephrased question, remember to fully express the information in the original question.

Original Question: Olivia has $23. She bought five bagels for $3 each. How much money does she have left?
Rephrased Question: Given a list of conditions, please answer the question. Condition 1: Olivia starts with $23. Condition 2: She buys five bagels, each costing $3. Question: How much money does Olivia have remaining after her purchase?

Original Question: Michael had 58 golf balls. On Tuesday, he lost 23 golf balls. On Wednesday, he lost 2 more. How many golf balls did he have at the end of Wednesday?
Rephrased Question: Given a list of conditions, please answer the question. Condition 1: Michael initially has 58 golf balls. Condition 2: On Tuesday, he loses 23 golf balls. Condition 3: On Wednesday, he loses 2 additional golf balls. Question: What is the total number of golf balls Michael has left at the end of Wednesday?

Original Question: Angelo and Melanie want to plan how many hours over the next week they should study together for their test next week. They have 2 chapters of their textbook to study and 4 worksheets to memorize. They figure out that they should dedicate 3 hours to each chapter of their textbook and 1.5 hours for each worksheet. If they plan to study no more than 4 hours each day, how many days should they plan to study total over the next week if they take a 10-minute break every hour, include 3 10-minute snack breaks each day, and 30 minutes for lunch each day?
Rephrased Question: Given a list of conditions, please answer the question. Condition 1: Angelo and Melanie need to study 2 textbook chapters and 4 worksheets. Condition 2: They allocate 3 hours per textbook chapter and 1.5 hours per worksheet. Condition 3: Their daily study limit is 4 hours, with a 10-minute break every hour, three 10-minute snack breaks, and a 30-minute lunch break each day. Question: Over the next week, for how many days should they plan to study to cover all their materials?

Original Question: Leah had 32 chocolates and her sister had 42. If they ate 35, how many pieces do they have left in total?
Rephrased Question: Given a list of conditions, please answer the question. Condition 1: Leah has 32 chocolates. Condition 2: Her sister has 42 chocolates. Condition 3: Together, they consume 35 chocolates. Question: How many chocolates remain between them after they have eaten some?

Original Question: There were nine computers in the server room. Five more computers were installed each day, from Monday to Thursday. How many computers are now in the server room?
Rephrased Question: Given a list of conditions, please answer the question. Condition 1: Initially, there are nine computers in the server room. Condition 2: Each day, from Monday to Thursday, five additional computers are installed. Question: What is the total number of computers in the server room after these installations?

Original Question: Jason had 20 lollipops. He gave Denny some lollipops. Now Jason has 12 lollipops. How many lollipops did Jason give to Denny?
Rephrased Question: Given a list of conditions, please answer the question. Condition 1: Jason starts with 20 lollipops. Condition 2: After giving some lollipops to Denny, Jason has 12 lollipops left. Question: How many lollipops did Jason give to Denny?

Original Question: Sam bought a dozen boxes, each with 30 highlighter pens inside, for $10 each box. He rearranged five of these boxes into packages of six highlighters each and sold them for $3 per package. He sold the rest of the highlighters separately at the rate of three pens for $2. How much profit did he make in total, in dollars?
Rephrased Question: Given a list of conditions, please answer the question. Condition 1: Sam purchases a dozen boxes of highlighters, with each box containing 30 pens, at $10 per box. Condition 2: He repackages five boxes into packages of six highlighters, selling each package for $3. Condition 3: He sells the remaining highlighters at a rate of three for $2. Question: What is Sam's total profit from these transactions?

Original Question: There are 15 trees in the grove. Grove workers will plant trees in the grove today. After they are done, there will be 21 trees. How many trees did the grove workers plant today?
Rephrased Question: Given a list of conditions, please answer the question. Condition 1: Initially, there are 15 trees in the grove. Condition 2: Grove workers will add more trees to the grove today. Condition 3: After planting, the total number of trees in the grove will increase to 21. Question: How many trees did the grove workers plant today?

**Original Question: {user question}**
**Rephrased Question:**

