# OpenReview forum: "Mutual Reasoning Makes Smaller LLMs Stronger Problem-Solver"
_ICLR.cc/2025/Conference — ICLR 2025 Poster_

### Official Review · Reviewer_LDgj · 2024-10-29

**Soundness:** 4
**Presentation:** 3
**Contribution:** 3
**Rating:** 8
**Confidence:** 4

**Summary:**

This paper tackles the challenge of improving small language models' reasoning abilities without fine-tuning or larger model supervision. The key innovation is a two-phase approach called rStar: first, a generator phase uses Monte Carlo Tree Search with an expanded set of reasoning operations (like decomposing problems, rephrasing questions, and proposing intermediate steps) to create potential solution paths. Then, a discriminator phase uses a second small language model to verify these solutions through "mutual consistency" (checking if the model can arrive at the same conclusion given partial steps). The approach is notable because it improves performance through better inference-time decision making rather than parameter updates or knowledge distillation from larger models. The empirical results demonstrate substantial improvements across multiple reasoning benchmarks and model sizes, suggesting that smaller language models have more latent reasoning capability than previously thought, but need better mechanisms to access it. The authors validate their approach through extensive ablation studies showing the importance of both the expanded MCTS action space and the mutual consistency verification.

**Strengths:**

The paper's primary originality lies in its novel two-phase approach to improving small language model reasoning. Rather than relying on conventional methods, it creatively combines an enriched Monte Carlo Tree Search with a mutual consistency verification phase using a second SLM. Particularly innovative is the expansion of the MCTS action space to include human-like reasoning operations such as problem decomposition, question rephrasing, and stepwise thinking. The proposed "mutual consistency" verification approach, while used before, is an interesting application that seems very effective.

The authors conduct a comprehensive evauation across a diverse set of benchmarks and models. Additionally, they perform convincing ablation studies that validate their key design choices, particularly the importance of both the expanded action space and mutual consistency verification. The provided baselines are also good.

The majority of the paper is well written and clearly expressed.

The significance of this work is particularly noteworthy. It demonstrates that SLMs possess stronger latent reasoning capabilities than previously believed and provides a practical method for improving SLM reasoning without requiring expensive fine-tuning or supervision from larger models.

**Weaknesses:**

The description of rStar in the introduction, starting on line 71, is hard to follow. Perhaps breaking the algorithm down into bullet points would help make the process more explicit and easier to digest than the wall of text. Moving figure 3 to the beginning would also be effective.

On line 255 you say "based on whether it reaches the correct answer" (which would imply you're using the ground truth answer during your search), but it seems like it is simply based on the likelihood of self-consistency majority voting mentioned on line 259.

It seems one potential limitation of mutual reasoning consistency is if an early action makes an incorrect statement that dramatically simplifies the problem. In this case it is likely that $SLM_2$ matches $SLM_1$. Given that this method works so well, this clearly isn't a critial issue, but certainly worth addressing/exploring more (at the very least mention this potential limitation).

Comparisons to other MCTS methods would be nice to have. A quick google search found (on top of the couple cited in the paper) "Monte Carlo Tree Search Boosts Reasoning via Iterative Preference Learning" and "Improve Mathematical Reasoning in Language Models by Automated Process Supervision". How do these approaches performance compare to yours?

**Questions:**

- When does performance improvement drop off with increased roll outs? The paper stops at 32, but performance seems to still be improving linearly.
- How does this approach compare to other MCTS methods recently proposed?

---

> ### Author Response · Authors · 2024-11-22
> **Official Comments by Authors**
>
> > ### When does performance improvement drop off with increased roll outs? The paper stops at 32, but performance seems to still be improving linearly.
>
> **Response**: Thank you for your question. We limited our experiments to 32 rollouts due to our constrained GPU resources, and we observed promising results at that level. Following your suggestion, we extended the experiments to 48 rolluts for LLaMA3-8b-Instruct. As shown below, increasing the rollouts to 48 can further improve the reasoning performance.
>
>
> |rollout|LLaMA3-8b-insutrct with rStar on GSM8K (%) |
> | :--:   |:--:   |
> |2|88.02|
> |4|89.16|
> |8|89.92|
> |16|90.14|
> |32|91.13|
> |**48**|**91.51**|
>
>
> > ### Comparisons to other MCTS methods would be nice to have. A quick google search found (on top of the couple cited in the paper) "Monte Carlo Tree Search Boosts Reasoning via Iterative Preference Learning" and "Improve Mathematical Reasoning in Language Models by Automated Process Supervision". How do these approaches performance compare to yours?
>
> **Response**: Thank you for your suggestion. Our rStar approach is complementary to many existing MCTS-based reasoning approaches. In addition to the two papers you recommended, we have also reviewed other recent representative works. MCTS-related research generally falls into the following categories:
>
> 1) Optimizing MCTS algorithms during LLM inference [1,3]:
> This includes methods like MCTSr and RAP, both of which we compare against or have discussed in our paper. For these approaches, our proposed diverse action space outperforms their single-type action space by enabling a broader exploration of potential solutions.
>
> 2) Training process reward models [2,5,6]:
> Examples include [5] and MindStar, which typically require expensive and challenging reward training data collection. These methods also face challenges in generalizing across different reasoning tasks. Our proposed mutual reasoning approach enables *general* effective solution verification without the need to train a dedicated reward model. Furthermore, mutual reasoning can complement reward models by guiding MCTS with diverse reward signals. This is something we plan to explore further in future work to strengthen the role of mutual reasoning.
>
> 3) Using MCTS to optimize LLM post-training[4]:
> Methods such as the first paper you mentioned [4] leverage MCTS to generate higher-quality solutions, which are then used to fine-tune LLMs for better Pass@1 reasoning accuracy. These approaches also propose novel preference learning algorithms for improved alignment. rStar does not involve fine-tuning LLMs. We're orthogonal to such methods.
>
> We appreciate your insightful comment, which has inspired us to carefully reflect on rStar's positioning and the immense potential of integrating it more closely with the recent MCTS approaches.
>
> [1] Reasoning with Language Model is Planning with World Model
>
> [2] MindStar: Enhancing Math Reasoning in Pre-trained LLMs at Inference Time
>
> [3] Accessing GPT-4 level Mathematical Olympiad Solutions via Monte Carlo Tree Self-refine with LLaMa-3 8B: A Technical Report
>
> [4] Monte Carlo Tree Search Boosts Reasoning via Iterative Preference Learning
>
> [5] Improve Mathematical Reasoning in Language Models by Automated Process Supervision
>
> [6] LLaMA-Berry: Pairwise Optimization for O1-like Olympiad-Level Mathematical Reasoning

---

> ### Author Response · Authors · 2024-11-22
> **Official Comments by Authors**
>
> > ### On line 255 you say "based on whether it reaches the correct answer" (which would imply you're using the ground truth answer during your search), but it seems like it is simply based on the likelihood of self-consistency majority voting mentioned on line 259.
>
> **Response**: Thank you for pointing this out. You are right, rStar does not have the groundtruth, i.e., "the correct answer", hence it utilizes the likelihood of self-consistency majority voting. We have clarified this in the updated paper.
>
>
> > ### It seems one potential limitation of mutual reasoning consistency is if an early action makes an incorrect statement that dramatically simplifies the problem. In this case it is likely that matches. Given that this method works so well, this clearly isn't a critical issue, but certainly worth addressing/exploring more (at the very least mention this potential limitation).
>
>  **Response**: Thank you for your insightful suggestion.  If an SLM makes an error in an earlier step, feeding the partial trace back to the same SLM as a hint can indeed lead to repeated mistakes. However, mutual reasoning consistency leverages a key insight: due to differences in training data and algorithms, the second discriminator SLM exhibits significant diversity compared to the reasoning model (i.e., the first generator SLM). As a result, in most cases, even if the partial trace contains mistakes, the second SLM typically produces a different incorrect answer. Then the generator SLM and the discriminator SLM would fail to reach agreement, which will be filtered out from final solution selection. But indeed, in rare cases where an early action makes an incorrect statement that dramatically simplies the problem, then the discriminator SLM may make identical errors. We have added a section on potential limitations in the updated paper.
>
>  We greatly appreciate your question, as it prompted us to further analyze our mutual reasoning traces. Interestingly, in some cases, even when the generator SLM makes errors in the earlier steps, the discriminator SLM identifies these mistakes and offers a corrected solution. An example is provided below:
>
>
>
>
> >Question: Josh decides to try flipping a house.  He buys a house for \\$80,000 and then puts in \\$50,000 in repairs.  This increased the value of the house by 150%.  How much profit did he make?
> >
> >groundtruth: 70,000
> >
>
> >**Error trace generated by LLaMA3-8B-instruct**: Let's think step by step. **The new value of the house is \\$80,000 + (150% * \\$80,000) = \\$80,000 + \\$120,000 = \\$200,000** (*wrong step, note that we feed this step along with the preceding steps as a partial trace to the discriminator SLM*). Josh bought the house for $80,000 and sold it for \\$200,000. His profit is \\$200,000 - \\$80,000 = \\$120,000. Now we can answer the question: Josh made a profit of \\$120,000. The answer is 120,000.
> >
>
> >**Correct solution#1 generated by discriminator SLM (phi3-mini-instruct)**:
> >But don't forget about the cost of repairs! To get his actual net profit, we subtract \\$50,000 (the repair costs) from \\$120,000 (profit). Therefore, Josh made a profit of \\$120,000 - \\$50,000 = \\$70,000. The answer is: 70000.
>
> >**Correct solution#2 generated by discriminator SLM (phi3-mini-instruct)**:
> >Also consider the cost of \\$50,000 repairs which makes his total investment $130,000 (\\$80,000 + \\$50,000). To calculate his net profit, we subtract his total investment from his selling price, \\$200,000 - \\$130,000 = \\$70,000. The answer is: 70000.

---

> > ### Comment · Reviewer_LDgj · 2024-11-25
> >
> > Thank you for addressing all of my questions. Very interesting example. It would be interesting to see how improving self-correction might improve this system in a future work.

---

> ### Author Response · Authors · 2024-11-25
> **Official Comment by Authors**
>
> Thank you for your very positive feedback and thoughtful comment! We are also excited about the potential of incorporating self-correction to improve rStar in future work.

---

### Official Review · Reviewer_hxm4 · 2024-11-03

**Soundness:** 3
**Presentation:** 3
**Contribution:** 3
**Rating:** 6
**Confidence:** 4

**Summary:**

The paper introduces rStar, a self-play mutual reasoning approach designed to improve the reasoning capabilities of small language models (SLMs). This method enhances SLMs with prompting engineering. The key mechanism involves a generation-discrimination process where the target SLM creates reasoning trajectories using Monte Carlo Tree Search (MCTS) enriched with human-like reasoning actions. Another SLM, with similar capabilities, acts as a discriminator to verify the generated trajectories, ensuring they are mutually consistent, which increases their likelihood of correctness. Experiments show that rStar effectively boosts performance on challenging reasoning benchmarks.

**Strengths:**

The idea of introducing human-like reasoning actions and using mutual consistency between SLMs to verify the results during inference time is intriguing. The writing is well-structured and clear. Empirical results with details are given.

**Weaknesses:**

There are still some design and implementation details related to manual action selection, self-reflection, token cost, scalability to larger models that need further clarification.

**Questions:**

1) The authors introduce a set of five human-like reasoning actions as the action types for MCTS. This design requires manual selection and experimental validation, which may not be optimal. Could the authors provide any experiments or analysis comparing their manually selected actions to other potential action sets? Did the authors consider automating the action type design process?


2) A very interesting aspect of OpenAI's o1 model is its self-reflective behavior. Did the authors consider integrating self-reflection as an action type in their framework? What potential benefits or challenges do the authors expect with such an addition?


3) The authors mention that temporal order constraints exist for different action types. Could the authors provide pseudocode or a more detailed explanation of how the temporal order constraints are implemented in their MCTS algorithm?


4) In Section A.3, the authors mention the high token cost associated with this method. For instance, the average number of generated tokens per question on GSM8k is 367.1k, which could limit the method's practical applicability. Have the authors considered optimization strategies to address this issue? While distributed inference can reduce processing time, it does not reduce the overall computational cost.


5) The authors emphasize that this method is designed for SLMs. Have the authors conducted experiments or analysis comparing rStar's performance on SLMs versus on LLMs? What’s the expectation of the method's effectiveness to change with model size?

---

> ### Author Response · Authors · 2024-11-22
> **Official Comments by Authors**
>
> >### The authors introduce a set of five human-like reasoning actions as the action types for MCTS. This design requires manual selection and experimental validation, which may not be optimal. Could the authors provide any experiments or analysis comparing their manually selected actions to other potential action sets? Did the authors consider automating the action type design process?
>
> **Response (1/2)**: Thank you for your insightful question. The set of five human-like reasoning actions was manually designed to help SLMs better generate correct solutions for challenging reasoning tasks.  We agree that automating the design of the action space is an interesting and valuable direction. Based on our current experience, such an approach would necessitate further improvements in SLM capabilities, such as the instruction-following, thus requiring significant training resources. We see this as a promising direction for our future work.
>
> 1) Regarding comparisons with other potential action spaces, we conducted a survey of recent MCTS-related papers and found that the exploration of action spaces remains relatively limited. As summarized in the table below, most methods rely on a single action type — either similar to our $A_1$ or $A_3$ - both of which are already included in our action space.  We also found that MCTSr[4] introduced a "self-refine" action, which iteratively polishes the generated solution. However, based on our experience, the "self-refine" action requires an instruction-tuned model and its effectiveness depends on model capabilities[8]. We see the potential of incorporating self-refine into our framework for stronger SLMs and plan to explore this in future work.
>
> || Action Space | Evaluated Model Size|
> | :--:   |:--:   | :--:   |
> |AlphaMath[1] |$A_1$: generate next step | 7B, finetune |
> |ToT[2] |$A_1$: generate next step | GPT4|
> |AlphaLLM [5]|$A_1$: generate next step | 70B, finetune |
> |ReST-MCTS* [6]|$A_1$: generate next step | 6B/7B, finetune |
> |MindStar [7]|$A_1$: generate next step | 7B/13B pretrained ckpt |
> |RAP [3]| $A_3$: propose a new sub-question along with its answer |LLaMA2-33B pretrained ckpt |
> |MCTSr [4]| $A_2$: generate all steps, self-refine | LLaMA3-8B-instruct|
> |**Ours**|$A_1$, $A_2$, $A_3$, $A_4$, $A_5$  |7B/8B, both pretrained ckpt and instruct version |
>
> We present an ablation study to evaluate the effectiveness of different action spaces, as shown in the table below. The experiments were conducted using LLaMA3-8B on 200 sampled GSM8K questions. The results indicate that each action in our proposed action space plays a critical role in enhancing reasoning accuracy. Compared to the commonly used action spaces in other works, which rely on a single action type (either $A_1$ or $A_3$), our five-action space significantly boosts accuracy.
>
> |Action Space|Accuracy (%)|
> | :--:   |:--:   |
> |$A_1$ | 35.5|
> |$A_3$ |70.5 |
> |$A_3$+$A_5$|72.5 |
> |$A_3$+$A_4$+$A_5$|73.5|
> |$A_2$+$A_3$+$A_4$+$A_5$|74.0|
> |$A_1$+$A_2$+$A_3$+$A_4$+$A_5$ (ours)|**75.0** |
>
> [1] AlphaMath Almost Zero: Process Supervision without Process https://arxiv.org/abs/2405.03553
>
> [2] Tree of thoughts: deliberate problem solving with large language models https://arxiv.org/abs/2305.10601
>
> [3] Reasoning with Language Model is Planning with World Model https://arxiv.org/abs/2305.14992
>
> [4] Accessing GPT-4 level Mathematical Olympiad Solutions via Monte Carlo Tree Self-refine with LLaMA3-8B   https://arxiv.org/abs/2406.07394
>
> [5] Toward Self-Improvement of LLMs via Imagination, Searching, and Criticizing https://arxiv.org/pdf/2404.12253
>
> [6] ReST-MCTS*: LLM Self-Training via Process Reward Guided Tree Search  https://arxiv.org/pdf/2406.03816
>
> [7] MindStar: Enhancing Math Reasoning in Pre-trained LLMs at Inference time https://arxiv.org/pdf/2405.16265
>
> [8] Analyzing the performance of self-refine on different large language models; https://github.com/anforsm/self-refine/blob/main/report.pdf

---

> ### Author Response · Authors · 2024-11-22
> **Official Comments by Authors**
>
> **Response (2/2)**:  2） **Case study of our action space**: In addition, we would like to present a case study demonstrating how our rich action space outperforms single action in augmenting SLMs for problem-solving. Below is LLaMA2-7B's response to a relatively challenging problem from GSM8K. As shown, using few-shot CoT or a single action such as $A_1$ or $A_3$ fails to solve the problem. Instead, leveraging a combination of different actions successfully yields the correct answer.
>
> >Question: Mike needed a new pair of jeans.  When he got to the mall he saw that his favorite jeans were advertised 25% off.  The original price of the jeans was \$40.  How much money will Mike have left over if he pays with a \50.00 bill?
> >
>
> > **Few-shot CoT (Wrong)**:  Let's think step by step. Mike needed a new pair of jeans.  When he got to the mall he saw that his favorite jeans were advertised 25% off.  The original price of the jeans was 40 dollars. If he pays with a 50 dollar bill, then he has 50 - 40 = 10 dollars. The answer is: 10.
> >
> >
>
> > **rStar with only $A_3$: subquestion-subanswer (Wrong)**:
> >
> > Node1: [$A_3$] What is the discounted price of the jeans?
> subanswer: The original price of the jeans is \\$40. The discount is 25%, so the discounted price is (1 - 0.25) * \\$40 = \\$30. The answer is 30.
> >
> >Node2: [$A_3$] How much money will Mike have left over if he pays with a \\$50.00 bill?
> subanswer: If he pays with a \$50 bill, he will have 30 - 50 = -\\$20 left over. The answer is -20.
> >
>
> >**rStar with only $A_1$: propose next one step (Wrong)**:
> >
> >Node1: [$A_1$] Mike needs a pair of jeans, but there is a discount so it costs 25% less than the original price of \\$40.
> >
> >Node2: [$A_1$] Since we know the original price, we can use this information and say that the discount is (\\$40 \u00d7 0.25) = 10 dollars.
> >
> >Node3: [$A_1$] The jeans cost 10 dollars less since the sale is 25%. So 50 - 10 is 40. This is the amount Mike spent.
> >
> >Node4: [$A_1$] The answer is: 40.
> >
>
> >**rStar trajectory-1 with diverse actions (Correct)**:
> >
> >Node1: [$A_1$] The original price is \$40.
> >
> >Node2: [$A_1$] The discount amount is 25%.
> >
> >Node3: [$A_2$]: Let's think step by step. Mike needed a new pair of jeans. His favorite jeans were originally priced at \\$40. When 25% off was applied, it became \\$30. Mike paid with a \$50 dollar bill, so he had \\$20 extra (because \\$50 - \\$30 = \\$20). The answer is: 20.
> >
>
> >**rStar trajectory-2 with diverse actions (Correct)**:
> >
> >Node1: [$A_3$]: How much money does Mike need to pay for a pair of jeans?
> >
> >subanswer: With a discount of 25%, the jeans cost \\$40 - (40 * 0.25) = \\$30. The answer is \\$30.
> >
> >Node2: [$A_2$]: Let's think step by step. Mike needs 30 dollars. Paying it with a 50 dollar bill leaves him with 20 dollars extra. The answer is: 20.

---

> ### Author Response · Authors · 2024-11-22
> **Official Comments by Authors**
>
> >### The authors mention that temporal order constraints exist for different action types. Could the authors provide pseudocode or a more detailed explanation of how the temporal order constraints are implemented in their MCTS algorithm?
>
> **Response**: Thank you for your question. While it is not strictly necessary to impose order constraints on the different action types for rStar to function, doing so helps avoid ineffective exploration and reduces the inference costs. We impose two simple order constraints: 1) A5 (rephrase the question) can only occur after the root question; and 2) A4 (answer the sub-question again) can only happen after A3 (propose next sub-question along with its answer). We have updated the paper to provide a clearer and more accurate explanation of these constraints. We hope this resolves your concerns, and we're happy to answer any further questions you may have.
>
> > ### A very interesting aspect of OpenAI's o1 model is its self-reflective behavior. Did the authors consider integrating self-reflection as an action type in their framework? What potential benefits or challenges do the authors expect with such an addition?
>
> **Response**: Thank you for your insightful suggestion. We have indeed consider integrating self-reflection/self-correction as an action type in our framework as part of future work. The main challenge lies in the self-reflection capability of the SLM itself, which would likely require substantial specialized fine-tuning to enable this functionality effectively. If the SLM can reliably perform self-reflection, we expect that integrating it into rStar can significantly improve reasoning performance. For example, after an MCTS rollout is completed, instead of directly performing a new rollout, self-reflection could assess whether the current rollout contained errors. If errors are detected, the model could learn from those errors and propose a new solution.
>
> > ### In Section A.3, the authors mention the high token cost associated with this method. For instance, the average number of generated tokens per question on GSM8k is 367.1k, which could limit the method's practical applicability. Have the authors considered optimization strategies to address this issue? While distributed inference can reduce processing time, it does not reduce the overall computational cost.
>
> **Response**: Thank you for your thoughtful comment. Indeed, while distributed inference can speed up inference time, it does not reduce the overall computational cost. However, we believe the practical applicability of rStar remains feasible, and we have identified several strategies to optimize its efficiency:
>
> 1) **Batch inference**. By increasing the batch size (i.e., performing MCTS rollouts for multiple problems simultaneously), we can improve GPU utilization and accelerate inference time, making rStar more efficient.
>
> 2) **Improving SLM capabilities**: We have observed that as the capabilities of the SLM continue to improve (as seen in recent trends), the model can achieve promising reasoning performance with fewer MCTS rollouts in rStar. This reduction in rollouts leads to a significant decrease in the number of generated tokens, thereby reducing computational costs.
>
>   3. **MCTS pruning**: We plan to incorporate pruning into the MCTS algorithm, which enables us to avoid or terminate ineffective explorations early. This can further reduce unnecessary token generation and overall computational overhead.
>
> > ### The authors emphasize that this method is designed for SLMs. Have the authors conducted experiments or analysis comparing rStar's performance on SLMs versus on LLMs? What’s the expectation of the method's effectiveness to change with model size?
>
> **Response**: Thank you for your insightful question. The primary reason we focused on applying rStar to SLMs (7B-8B) was due to the limitations of our available GPU resources. While rStar can indeed be applied to larger model sizes, doing so would require more computational resources.
>
> To demonstrate the effectiveness of rStar, we conduct experiments using a 12B LLM (Mistral-Nemo-Instruct-12B) with 32 rollouts. The results on GSM8K dataset are as follows. We can see that rStar remains effective when scaled to larger LLMs, demonstrating its potential for broader applicability.
>
> |Method|Mistral-Nemo-Instruct (12B) GSM8K Accuracy (%)|
> | :--:   |:--:   |
> |Few-shot CoT| 75.8|
> |SC (8)| 84.2|
> |SC (32)| 86.8|
> |SC (128)| 87.1|
> |**rStar (32 rollouts)**| **91.1**|

---

> > ### Comment · Reviewer_hxm4 · 2024-12-03
> > **Official Comment by Reviewer hxm4**
> >
> > Thank you for addressing my questions. I believe there are many interesting directions to explore as potential future work.
> > Also, prompting alone may have limitations in enhancing the capabilities of SLMs. Thus, integrating fine-tuning might be necessary to achieve significant improvements on complex reasoning tasks such as the MATH benchmark.

---

> ### Author Response · Authors · 2024-12-03
> **Official Comment by Authors**
>
> Thank you for your insightful feedback and suggestions. We are actively exploring fine-tuning-based approaches, and our preliminary results indicate promising improvements. We look forward to sharing more insights and findings in future extensions of rStar.

---

### Official Review · Reviewer_Uyvd · 2024-11-03

**Soundness:** 2
**Presentation:** 3
**Contribution:** 3
**Rating:** 6
**Confidence:** 3

**Summary:**

The paper describes a method to improve "reasoning capabilities" of small language models. This is done by generating trees of prompt using MCTS and by using a second small language model that "critiques" MCTS rollouts. After introducing the method the paper than show in its experimental section that on a suite of benchmarks the new method, dubbed rStar, outperforms existing methods.

**Strengths:**

- The paper is relatively well written making it easy to follow..
-  the methodology is clearly explained and differences as well as similarities to competing methods are made explicit.
- In the experimental evaluation it is clearly shown that the proposed method out-competes state-of-the-art approaches. This is substantiated by a couple of informative ablation studies.

**Weaknesses:**

The paper claims to improve reasoning capabilities of small language models. However, by training/augmenting a SML with MCTS on a specific dataset we now end up with a model that is informed by the statistics of the dataset. In the end there is no reasoning happening but the proposed method allows the SML to better exploit statistical patterns in the data-query pairs that it was trained on.

In the end the proposed methods does not allow SMLs to reason but is a method to perform prompt-engineering in an automated fashion using the statistics of the benchmark in question.


One of the problems I have is the anthropomorphism present in the paper. Specifically, with using "rich set of human like reasoning actions" already in the abstract and continuing it throughout the paper.


While I do not see how the paper enables SMLs to reason better, I can see that the introduces techniques have clear experimental advantages over competing methods. This leads me to tend towards arguing towards accepting the paper ever so slightly. .

**Questions:**

Reasoning capabilities are often studied in terms of generalizability How would you study generalization capabilities of your method? Could you transfer between tasks/benachmarks?

---

> ### Author Response · Authors · 2024-11-22
> **Official Comments by Authors**
>
> > ### The paper claims to improve reasoning capabilities of small language models. However, by training/augmenting a SML with MCTS on a specific dataset we now end up with a model that is informed by the statistics of the dataset. In the end there is no reasoning happening but the proposed method allows the SML to better exploit statistical patterns in the data-query pairs that it was trained on. Reasoning capabilities are often studied in terms of generalizability How would you study generalization capabilities of your method?  Could you transfer between tasks/benchmarks?
>
> **Response**: Thank you for your thoughtful review and for providing a positive rating for our paper. We truly appreciate the time and effort you dedicated to evaluating our work.
>
>  We would like to **clarify that our method does NOT involve training or fine-tuning an SLM with MCTS on a specific dataset**. Instead, the SLM remains fixed throughout the process, and MCTS is applied at inference time as a reasoning framework to help the SLM to generate higher-quality solutions. To address your concerns regarding reasoning and generalization, we provide the following clarifications:
>
> 1) **rStar generalizes well across different reasoning tasks**. When a new task or benchmark is introduced, rStar does not require much domain-specific knowledge. Only 1-2 few-shot examples are required. In our experiments, we used GPT-4 to write a few task-specific demonstrations for each action across five reasoning tasks. As shown in Table 2 of the original paper, we demonstrate strong generalization across diverse math and general reasoning tasks. To further highlight rStar's generalization effectiveness, we evaluate it on an additional non-math general reasoning task, FOLIO[1]. As shown in the following table, rStar significantly improves SLMs' accuracy.
>
> |Method| LLaMA3-8B| LLaMA3-8B-Instruct|
> | :--:   |:--:   | :--:   |
> |Few-shot CoT| 53.20| 58.62|
> |SC (8)| 55.17| 61.08|
> |SC (64)| 58.62| 61.08|
> |SC (128) |60.10|61.58 |
> |RAP (32 rollouts)|60.01|54.68|
> |**rStar (32 rollouts)**|**65.52** |**69.46** |
>
> [1] https://arxiv.org/abs/2209.00840
>
> 2) **rStar enables SLMs to better reason on unseen/untrained challenging math benchmarks**. To show that rStar truly enhances the reasoning capabilities of SLMs, rather than just allowing them to better exploit statistical patterns from previously seen data, we test it on 22 problems from the **AMC 2024**, which were released in January 2024. Since the SLMs (Mistral and LLaMA3-8B) were trained on data available before December 2023[2,3], there is no data leakage on the AMC 2024. As shown in the table below, rStar substantially improves the two SLMs performance on the challenging AMC 2024 benchmark.
>
> |Method|Mistral-7B-v0.1 (knowledge cutoff: before October 2023) |  LLaMA3.1-8B-Instruct (knowledge cutoff: December 2023)|
> | :--:   |:--:   | :--:   |
> |Few-shot CoT| 22.72% |  18.18%|
> |SC (128)| 18.18% | 31.82%|
> |**rStar (32 rollouts)**| **31.82%**| **40.91%**|
>
>
> [2] https://github.com/meta-llama/llama-models/blob/main/models/llama3_1/MODEL_CARD.md
>
> [3] https://arxiv.org/pdf/2310.06825
>
> 3) **Clarfications of rStar's key insights**: To provide a clearer explanation of our methodology, we would like to highlight three key insights that enables better reasoning: 1) *Scaling test-time computation*: our method coincidentally aligns with the recent scaling test-time computation advocated by GPT-o1, which shows that generating more tokens during inference can improve LLM's performance. Unlike traditional approaches where SLMs/LLMs attempt to solve reasoning tasks in a single inference pass, we decouple reasoning into two stages: solution generation and verification. During solution generation, MCTS augments the SLM to explore **multiple candidate solutions**, while the verification stage selects the (more likely) correct solution from these candidates. While this increases inference cost, it significantly boosts reasoning performance. 2) *Step-by-step generation*: Unlike prior methods where LLMs generate an entire solution in one inference, we use MCTS with diverse action spaces to guide the SLM in generating one reasoning step at a time. This decomposes the end-to-end reasoning task into smaller and easier subtasks, making it more manageable for the SLM and leading to higher-quality solution generation; 3) *General and effective solution selection through mutual consistency*: Instead of training task-specific reward models for answer verification, our method of using another SLM for mutual consistency inherently leads to better generalization. As a result, we consistently achieve performance gains across diverse reasoning tasks, as shown in Table 2 (see the original paper).
>
>
> We hope this explanation addresses your questions and provides greater clarity to our approach. Thank you again for your valuable feedback, and we welcome any additional questions or suggestions you might have!

---

### Official Review · Reviewer_AiN4 · 2024-11-04

**Soundness:** 2
**Presentation:** 3
**Contribution:** 3
**Rating:** 5
**Confidence:** 2

**Summary:**

The paper puts forward an approach to improve reasoning capabilities of LLMs. The approach is based on MCTS with a discriminator model selecting the most promising answers. Experiments appear to demonstrate superior reasoning capabilities of the approach compared to fine-tuning and other approaches.

Post-rebuttal: My expertise in the area is limited to the one of an interested observer. I am reassured by the responses of the authors to my questions and having read through the related links and explanations, I am happy to revise my score up and recommend acceptance. I would encourage the authors to add explanations to points such as the ones I have highlighted to increase the reach of the paper to people outside the LLM area.

**Strengths:**

The experimental results point to much improved performance of the model under reasoning.

**Weaknesses:**

This is not my area, hence low confidence, however as someone outside the LLM area I struggled to see the arguments put forward in the paper. I recommend the author to put more emphasis into making the key parts of the technical material more accessible by providing more explanations. Here are the key problems I had.

1 I did not understand how the MCTS approach works here both in terms of generation of the next steps. What simulation is performed, how this is tuned and where the MCTS itself results from (is it the model itself). Equally I found no explanation as to why this would/should lead to increased performance in the step generations.

2 I did not understand why a discriminator would have better performance than the original model and why it is the case that these capabilities cannot or should not be employed directly by the reasoning model.

I think that if I had more explanations on the points above I would have been to understand the technical contribution more.

**Questions:**

Answering the questions 1, 2 above would help.

Edit - post rebuttal: questions 1 and 2 were answered by the authors to my satisfaction.

---

> ### Author Response · Authors · 2024-11-22
> **Rebuttal by Authors**
>
> Thank you for taking the time to review our work, especially given that it falls outside your primary area of expertise. We'd like to provide more detailed explanations to address your questions and hopefully clarify our key contributions.
>
> >### I did not understand how the MCTS approach works here both in terms of generation of the next steps. What simulation is performed, how this is tuned and where the MCTS itself results from (is it the model itself). Equally I found no explanation as to why this would/should lead to increased performance in the step generations.
>
> **Response**: Thank you for your question. We didn't provide a detailed explanation in the main text because applying MCTS for next-step generations is a well-known approach; it has been used to improve LLM reasoning capabilities[1,2,3]. Our goal was to highlight our key insights and core technical contributions without detracting from the main focus. That said,  we're happy to provide a more detailed explanation here and address any further questions you may have.
>
> 1) Specifically, we start at the root node (i.e., the given question) and treat the entire trajectory as the current state (with the initial state being the question). LLM is then prompted to generate the next step, $s_i$, based on this state. If the LLM generates the next step directly, the results would be similar to greedy decoding with comparable reasoning performance. Instead, in the MCTS approach, the LLM is prompted to generate **multiple candidate nodes** for each predefined action type, as detailed in Section 3.2. MCTS then **selects the optimal response** for step $s_i$ based on the UCT score. If the UCT score is accurate, this approach can significantly improves the reasoning performance by finding higher-quality LLM responses for each reasoning step. The UCT formula is as follows:
>
>     $UCT(s, a) = \frac{Q(s, a)}{N(s, a)} + c \sqrt{\frac{\ln N_{parent}(s)}{N(s, a)}}$
>
>
> 2) The effectiveness of the MCTS approach relies heavily on the accruacy of Q-value and UCT score. The simulation process is used to iteratively update the Q-value and UCT score for each node, which is achieved through the standard rollout policy. Initially, the Q-values of all candidate nodes are set to 0, leading MCTS to randomly select a node for each step generation. This process continues until a terminal node is reached, which provides a final answer to the question. If the terminal node's answer is correct, its Q-value is set to 1, and this value is back-propagated along the trajectory to update the Q-values of all nodes in the path. Over time, as more rollouts are performed, MCTS becomes less random and increasingly effective. It graudally learns to select the highest-quality candidate node for each reasoning step, improving its overall decision-making process.
>
> [1] https://arxiv.org/abs/2305.14992, emnlp 2023
>
> [2] https://arxiv.org/abs/2405.03553, neurips 2024
>
> [3] https://arxiv.org/abs/2408.03314, google deepmind

---

> ### Author Response · Authors · 2024-11-22
> **Rebuttal by Authors**
>
> >### I did not understand why a discriminator would have better performance than the original model and why it is the case that these capabilities cannot or should not be employed directly by the reasoning model.
>
> **Response**: Thank you for your question. The reasoning model (the target SLM) is designed to generate potential solutions, not to verify them. The discriminator (another SLM) provides mutual verification for the candidate solutions generated by the reasoning model. This design choice was made for two key reasons.
>
> 1) **Limited Self-Evaluation:** The SLM's limited capabilities hinder its ability to perform effective self-verification and reliably select the correct solution from candidates. Self-evaluation for scoring each candidate node often yields near-to-random results. To illustrate, in an ablation study on RAP[1] (a representative MCTS approach, and serves as our baseline) with Mistral as the generator (see Table 6 in Appendix A.1, we also show the numbers in the table below), replacing the self-evaluated $r_1$ score with random scores showed no significant impact on final performance. This suggests that **SLM performs near-random self-evaluation during the solution generation**.
>
> |Method| Mistral (%)|
> | :--:   |:--:   |
> | RAP  | 56.25|
> | RAP + random $r_1$ score| 55.50|
>
> 2) **Mutual consistency to reach agreement on answer**: We therefore advocate using another SLM for mutual reasoning consistency, based on two key insights: (i) Due to differences in training data and algorithms, the second SLM exhibit significant diversity compared to the reasoning model (generator SLM). This diversity typically leads to diverse responses from each model for the same question. If both models agree on an answer, it is more likely to be correct. Notably, we have found that it is rare for two different SLMs to provide identical incorrect answers.  (ii) Instead of fully relying on the second SLM for solution selection, we use it only to provide an answer for cross-validation. An answer is retained only when both SLMs agree, which we called as mutual reasoning consistency. Our empirical experimental results, which significantly outperform other baseline methods, prove the effectiveness of mutual consistency.
>
>
> We hope our clarifications have provided a clearer explanation of the key insights and contributions of our approach. Given the novelty and demonstrated effectiveness of this approach, we believe in the value of our work. We welcome any further questions or suggestions you may have. Thank you again for your time and consideration.
>
> [1] https://arxiv.org/abs/2305.14992, emnlp 2023

---

### Meta-Review · Area_Chair_BcbH · 2024-12-22

**Metareview:**

(a) Summary:
The paper introduces rStar, a novel self-play mutual reasoning approach that improves small language models' (SLMs) reasoning capabilities without requiring fine-tuning or supervision from larger models. The key technical contributions include:
1. A generation-discrimination process where the target SLM uses MCTS augmented with human-like reasoning actions to generate solutions
2. A verification mechanism using another SLM as a discriminator to ensure mutual consistency
3. Extensive empirical validation showing significant improvements across multiple reasoning benchmarks (GSM8K, GSM-Hard, MATH, SVAMP, StrategyQA)

(b) Strengths:
1. Novel Technical Approach: The combination of MCTS with diverse reasoning actions and mutual consistency verification is innovative and well-justified
2. Strong Empirical Results: Comprehensive evaluation across multiple benchmarks and model sizes, with significant performance improvements
3. No Fine-tuning Required: Method works with pretrained models, making it widely accessible
4. Thorough Ablation Studies: Clear demonstration of the importance of each component
5. Good Generalization: Shows effectiveness across different reasoning tasks and on unseen problems

(c) Weaknesses:
1. High Computational Cost: The method requires substantial token generation
2. Manual Action Selection: The five reasoning actions are manually designed, raising questions about optimality
3. Limited Model Size Exploration: Primary focus on 7B-8B parameter models due to computational constraints
4. Potential Edge Cases: In rare cases, early incorrect statements might lead to consistent but wrong answers

(d) Reasons for Acceptance:
1. Significant Technical Innovation: The paper introduces a novel approach combining MCTS, diverse reasoning actions, and mutual consistency verification
2. Strong Empirical Results: Demonstrates substantial improvements across multiple benchmarks
3. Practical Impact: Method works with pretrained models and doesn't require fine-tuning or larger model supervision
4. Thorough Evaluation: Comprehensive ablation studies and analysis validate the approach
5. Clear Presentation: Well-written with clear methodology explanation

**Additional Comments On Reviewer Discussion:**

The discussion period featured four reviewers with scores ranging from 5 to 8. Reviewer AiN4 initially struggled with understanding the MCTS implementation and discriminator justification, but increased their score from 3 to 5 after receiving detailed technical explanations from the authors. Reviewer Uyvd (score: 6) questioned whether true reasoning was occurring, leading the authors to demonstrate generalization to unseen problems. Reviewer hxm4 (score: 6) raised concerns about implementation details and computational costs, which the authors addressed through ablation studies and optimization strategies. Reviewer LDgj gave the highest score (8) and requested additional comparisons, which the authors provided through extended experiments.
Throughout the rebuttal phase, the authors effectively addressed all concerns by providing detailed technical explanations, demonstrating generalization capabilities, and showing comprehensive experimental results. Their thorough responses strengthened the paper's contribution and supported its acceptance at ICLR 2025.

---

### Decision · Program_Chairs · 2025-01-22

Accept (Poster)